# The hexosamine pathway and coat complex II promote malignant adaptation to nutrient scarcity

Helena Dragic[1], Audrey Barthelaix[2], Cédric Duret[1] , Simon Le Goupil[3] , Hadrien Laprade[3], Sophie Martin[3], Sabine Brugière[4], Yohann Couté[4] , Christelle Machon[1,5], Jerome Guitton[1,5] , Justine Rudewicz[6], Paul Hofman[7], Serge Lebecque[1], Cedric Chaveroux[1] , Carole Ferraro-Peyret[1,8] , Toufic Renno[1], Serge N Manié[1,3]

The glucose-requiring hexosamine biosynthetic pathway (HBP), which produces UDP-N-acetylglucosamine for glycosylation reactions, promotes lung adenocarcinoma (LUAD) progression. However, lung tumor cells often reside in low-nutrient micro-environments, and whether the HBP is involved in the adaptation of LUAD to nutrient stress is unknown. Here, we show that the HBP and the coat complex II (COPII) play a key role in cell survival during glucose shortage. HBP up-regulation withstood low glucose-induced production of proteins bearing truncated *N*-glycans, in the endoplasmic reticulum. This function for the HBP, alongside COPII up-regulation, rescued cell surface expression of a subset of glycoproteins. Those included the epidermal growth factor receptor (EGFR), allowing an EGFR-dependent cell survival under low glucose in anchorage-independent growth. Accordingly, high expression of the HBP rate-limiting enzyme GFAT1 was associated with wild-type EGFR activation in LUAD patient samples. Notably, HBP and COPII up-regulation distinguished LUAD from the lung squamous-cell carcinoma subtype, thus uncovering adaptive mechanisms of LUAD to their harsh microenvironment.

## Introduction

Lung cancer remains the leading cause of cancer-related deaths worldwide (Pennell et al, 2019). About 85% of lung cancers are non–small-cell lung carcinoma (NSCLC), the most common subtypes being lung adenocarcinoma (LUAD) and lung squamous-cell carcinoma (LUSC). NSCLC displays a high rate of glucose consumption

to support anabolic pathways. Paradoxically, the high rate of glucose consumption by tumor cells and the poor vascular supply in certain parts of tumors (Hensley et al, 2016) result in a lower average concentration of glucose in the lung tumor microenvironment than in normal tissue (Urasaki et al, 2012; Wikoff et al, 2015; Hensley et al, 2016). Thus, distinct cluster of tumor cells likely exist within lung tumors with differential glucose access depending on the proximity of blood vessels.

Glucose is a main fuel for the hexosamine biosynthetic pathway (HBP) that produces UDP–N-acetylglucosamine (UDP-GlcNAc) for protein glycosylation (Akella et al, 2019). Accordingly, lowering glucose concentration in cell culture media often reduces cellular UDP-GlcNAc levels (Wellen et al, 2010; Abdel Rahman et al, 2013). In mice with impaired HBP flux, low UDP-GlcNAc translates into severely decreased *O*-GlcNAc modifications (*O*-GlcNAcylations) (Boehmelt et al, 2000), a tumor-promoting post-translational modification consisting in the addition of a single GlcNAc residue to specific serine/threonine residues of proteins (Akella et al, 2019). In contrast to *O*-GlcNAcylations, *N*-linked protein glycosylation that uses ~5% of the cellular UDP-GlcNAc pool is only partly affected by low UDP-GlcNAc levels (Boehmelt et al, 2000). *N*-glycan assembly at the ER membrane begins with two GlcNAc residues that are elongated with nine mannoses and three glucoses as an *N*-glycan precursor attached to dolichol (dolichol-linked oligosaccharide or DLO). Glucose is also a major carbon source for de novo synthesis of the nucleotide sugars UDP-glucose and GDP-mannose acting as building blocks of the *N*-glycan precursor biosynthesis (Nakajima et al, 2010; Harada et al, 2013). The core glycan is then transferred "en bloc" to asparagine residues of nascent proteins in the ER lumen and its trimming regulates productive protein folding (Stanley et al, 2015). Once glycoproteins are transported toward the Golgi by the coat protein II

[1]Centre de Recherche en Cancérologie de Lyon, INSERM U1052, Centre National de la Recherche Scientifique (CNRS) 5286, Centre Léon Bérard, Université de Lyon, Université Claude Bernard Lyon 1, Lyon, France  [2]Institute for Regenerative Medicine and Biotherapy (IRBM), Université de Montpellier, INSERM, Montpellier, France  [3]Inserm U1242, Centre de Lutte Contre le Cancer Eugène Marquis, Université de Rennes, Rennes, France  [4]Université Grenoble Alpes, INSERM, Commissariat à l'Energie Atomique (CEA), Unite Mixte de Recherche (UMR) BioSanté U1292, CNRS, CEA, FR2048, Grenoble, France  [5]U Hospices Civils of Lyon, Biochemistry and Pharmaco-toxicology Laboratory, Lyon Sud Hospital, Lyon, France  [6]Bordeaux Bioinformatics Center, CBiB, University of Bordeaux, Bordeaux, France  [7]Laboratory of Clinical and Experimental Pathology, Federation Hospitalo-Universitaire (FHU) OncoAge and BB-0033-00025, Nice University Hospital, IRCAN Antoine Lacassagne Center, Côte d'Azur University, Nice, France  [8]Hospices Civils de Lyon, Biopathology of Tumours, GHE Hospital, Bron, France

Correspondence: s.manie@rennes.unicancer.fr
To the memory of Sofiane Belfeki.

(COPII) complex, *N*-glycans are further modified by subsequent addition of different sugars, including GlcNAc, to generate complex *N*-glycans (Moremen et al, 2012). Thus, UDP-GlcNAc usage for *N*-glycosylation occurs both in the ER and the Golgi (Stanley et al, 2015). Whereas UDP-GlcNAc shortage results in altered Golgi *N*-glycan–branching (Boehmelt et al, 2000; Lau et al, 2007; Wellen et al, 2010), previous studies on ER-located DLO biosynthesis suggested that the nucleotide sugar GDP-mannose rather that UDP-GlcNAc is the limiting molecule in glucose-deprived cells for core glycan building. This mannose limitation results in defective DLO species that are rapidly degraded, leading to impaired protein glycosylation (Nakajima et al, 2010; Harada et al, 2013).

Even though a role for the HBP in LUAD progression was recently reported (Taparra et al, 2018; Kim et al, 2020), we still do not understand how these tumoral cells adapt to the native low level of glucose that impacts *N*-glycan production and UDP-GlcNAc–dependent tumor progression.

Here, we show that upon glucose limitation, truncated *N*-glycans were attached to newly synthesized proteins in the ER, highlighting a leakage in the quality control processes of *N*-glycosylation. Concomitant HBP up-regulation relieved a mannose deficiency in their elongation, rather than a deficiency in GlcNAc addition. This unforeseen function of HBP in rescuing *N*-glycan building in the ER, alongside a distinct increase in COPII components, promoted a wild-type epidermal growth factor receptor (EGFR)–dependent cell survival, which was specific to the LUAD subtype.

## Results

### HBP up-regulation is higher in human LUAD than in LUSC

To examine the status of the HBP (Fig 1A) in human NSCLC, we assessed the expression level of its rate-limiting enzyme paralogs *GFPT1* and *GFPT2* using publicly available gene expression data of paired tumor-normal samples (LUAD n = 58; LUSC n = 52) from The Cancer Genome Atlas (TCGA). We found that the mRNA levels of *GFPT1* were higher in LUAD than in LUSC tumor tissues compared with adjacent normal tissue and that *GFTPT2* expression levels did not differ statistically (Fig 1B). In line with its gene expression, the amount of GFAT1 protein (encoded by the *GFPT1* gene) tended to be more abundant in LUAD than in LUSC, as shown by immunoblotting of paired nontumoral and tumoral samples (Fig 1C and Table S1). These results suggested a stronger activation of the HBP in LUAD than in LUSC. To verify this, we examined the steady-state levels of the HBP metabolites N-acetylglucosamine 6-phosphate (GlcNAc-6P) and of UDP-N-acetylhexosamines (UDP-HexNAc) in paired LUAD and LUSC samples. UDP-HexNAc is the pool of UDP-GlcNAc and UDP-N-acetylgalactosamine (UDP-GalNAc), which exist in a dynamic equilibrium and cannot be discriminated by targeted liquid chromatography–mass spectrometry (LC–MS/MS). Fig 1D shows that GlcNAc-6P and UDP-HexNAc were more abundant in LUAD than in LUSC tumor tissues. Taken together, these results reveal a higher up-regulation of the HBP in the LUAD subtype of human NSCLC.

### UDP-HexNAc levels are less affected than those of other nucleotide sugars in transformed HBECs facing low glucose availability

Next, we used immortalized human bronchial epithelial cells (HBECs) that have been transformed through the combination of p53 knockdown and the exogenous expression of the *KRas^V12* oncogene, mimicking common genomic alterations in LUAD (Sato et al, 2013). Because the nutrient-rich composition of traditional culture media does not reflect the limitation in glucose in the lung tumor microenvironment, transformed HBECs that consumed 5–6 mM glucose/24 h under our experimental settings (Fig S1A) were maintained in a daily refreshed medium containing sufficient (10 mM) or limited (1, 0.1, or 0 mM) amounts of glucose for 48 h. Glucose shortage to 1 or 0.1 mM resulted in reduced cellular proliferation (Fig 2A) but, in contrast to complete glucose deprivation (0 mM), did not induce cell death (Fig S1B). The complete glucose deprivation condition was not further considered to avoid excessive cellular stress. In line with previous findings, low glucose increased the mRNA expression of *GFPT1* and the protein level of GFAT1 (Chaveroux et al, 2016; Moloughney et al, 2016) (Fig 2B). Of note, mRNA levels of GFPT2, the paralog of GFPT1, were only slightly expressed in HBECs (Fig S1C).

Then, we investigated the consequences of glucose shortage on nucleotide sugars composing core glycans in the ER, the cellular levels of which are regulated by glucose availability (Harada et al, 2013), namely UDP-GlcNAc, GDP-mannose, and UDP-glucose. GDP-galactose, GDP-glucose, and GDP-mannose cannot be discriminated by LC–MS/MS and were collectively referred to as GDP-hexose. As expected, all nucleotide sugars were less abundant in transformed HBECs after glucose shortage (0.1 mM) for 48 h (Fig 2C). However, UDP-HexNAc levels were less affected than that of GDP-hexose or UDP-glucose. These results are in line with previous findings showing that glucose deprivation results in depletion of intracellular pool of GDP-mannose rather than that of UDP-GlcNAc (Nakajima et al, 2010; Harada et al, 2013). Indeed, in contrast to *GFPT1*, transcripts of phosphomannose isomerase (*MPI*) and phosphomannomutase (*PMM2*) enzymes of the mannose pathway (Fig S1D) were down-regulated in glucose-limited conditions (Fig S1E). Thus, in transformed HBECs, low glucose levels lead to a reduction in GDP-hexose and UDP-glucose availability, whereas that of UDP-HexNAc are preserved.

### Preserved levels of UDP-GlcNAc upon glucose shortage sustain the *N*-glycosylation pathway in the ER rather than the *O*-GlcNAcylation pathway

To extend the above findings, we assessed toward which of the two *O*-GlcNAcylation and *N*-glycosylation pathways residual UDP-GlcNAc was assigned to (Fig 1A). Western blot analyses showed that growing cells in 0.1 mM glucose for 48 h strongly reduced the cellular pattern of *O*-GlcNAcylation (Fig 3A, upper panel, RL2 antibody), indicating that this pathway is highly susceptible to reduced levels of UDP-GlcNAc in HBECs. Conversely, we observed an increase in *N*-GlcNAc2–modified proteins (Fig 3A, lower panel, CTD110.6 antibody). These latter proteins were described to be ER-located proteins bearing truncated *N*-glycans (*N*-linked GlcNAc2 structures, Fig 3B) that are not further elongated by mannose

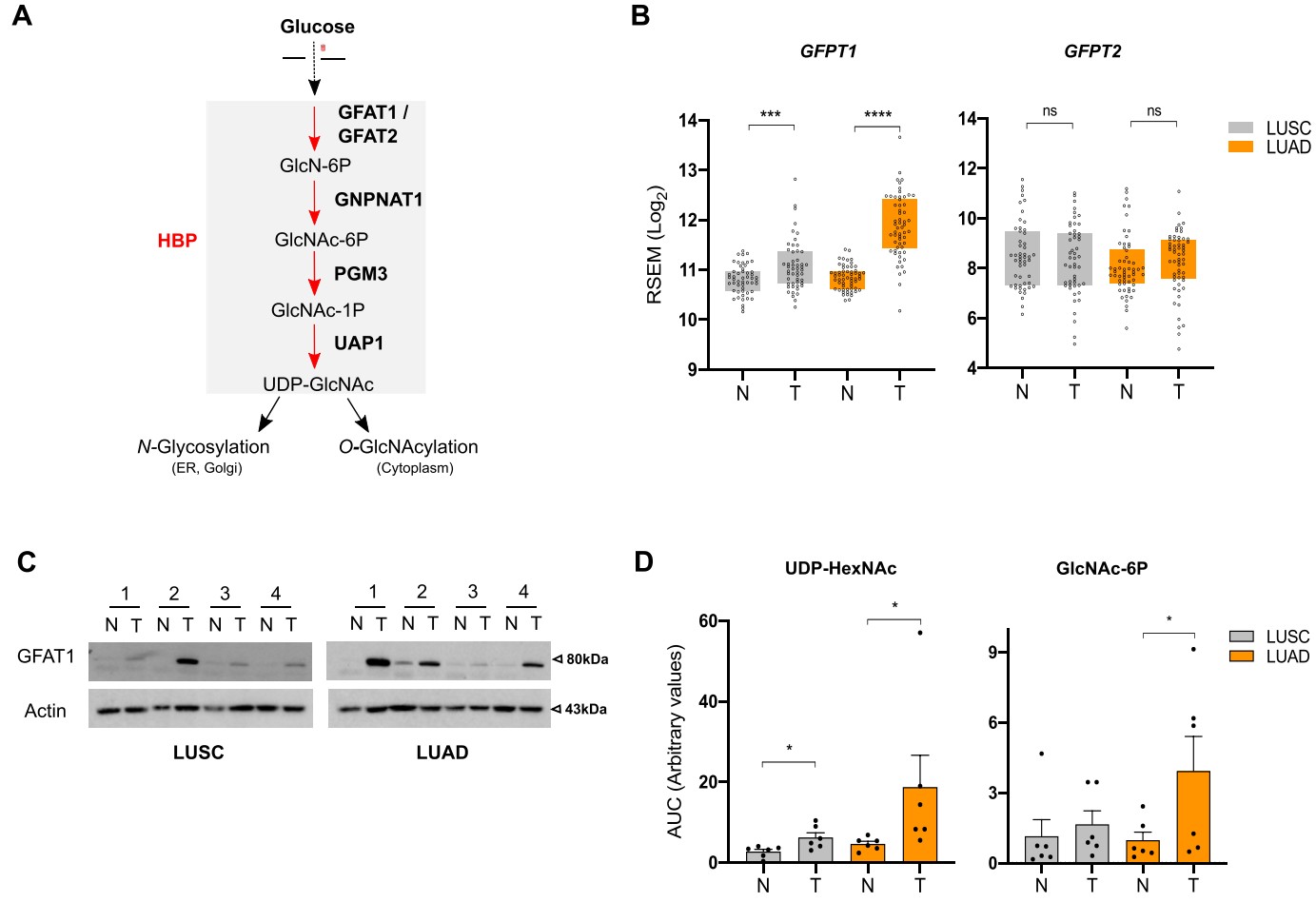

**Figure 1.  Human LUAD display distinctive hexosamine biosynthetic pathway (HBP) up-regulation.**
**(A)** Schematic representation of the HBP. **(B)** The Cancer Genome Atlas analysis of HBP enzymes in LUAD and LUSC paired nontumor (N) and tumor (T) tissue samples, presented as Log$_2$-transformed RSEM-normalized count. ****$P$ < 0.0001, **$P$ < 0.01, ns, nonsignificant (Mann–Whitney). **(C)** Western blotting of GFAT1 in LUAD and LUSC paired nontumor (N) and tumor (T) tissue samples. Actin is used as a loading control. **(D)** Abundance of UDP-HexNAc and GlcNAc-6P in LUAD and LUSC paired nontumor (N) and tumor (T) tissue samples, presented as the ratio of the area under the curve (AUC) of compound of interest/AUC of internal standard. *$P$ < 0.05 (Wilcoxson test, n = 6 independent experiments, mean ± SEM).
Source data are available for this figure.

residues upon glucose deprivation and which cross-react with the anti-*O*-GlcNAc CTD110.6 antibody (Isono, 2011; Isono et al, 2013; Reeves et al, 2014; Sasaoka et al, 2018). In agreement with this, Fig 3A shows that the binding of CTD110.6 was sensitive to the endoglycosidase PNGase (P), removing all *N*-linked carbohydrates, but not to Endo H (E) that cleaves N-linked structures only if branched with mannose residues (Trimble & Tarentino, 1991; Freeze & Kranz, 2010) (Fig 3B). Treatment of HBECs for the last 24 h of culture with the nucleoside antibiotic tunicamycin, an inhibitor of *N*-linked glycosylation of proteins by competing with UDP-GlcNAc for active site binding (Keller et al, 1979), prevented the appearance of *N*-GlcNAc2 structures (Fig 3C). These results suggested that the increase in *N*-GlcNAc2 structures under low-glucose condition were because of a dynamic fueling of the *N*-glycosylation pathway rather than the accumulation of a cleavage product of mature *N*-glycosylated proteins. In line with this, low glucose-induced *N*-linked GlcNAc2 structures have been detected on newly synthesized proteins in the ER (Sasaoka et al, 2018).

Together, these data show that the residual amount of UDP-GlcNAc after glucose shortage is actively channeled into ER-located *N*-glycosylation, leading to the production of proteins bearing truncated *N*-glycans.

We next tested whether *N*-GlcNAc2–modified proteins could also be detected in tumor tissue samples. We found variable pattern of *O*-GlcNAcylation (RL2 signal intensity) in tumor tissues compared with adjacent normal tissue (Fig S2). The difference in *O*-GlcNAcylation levels in a paired sample correlated with the respective amount of the *O*-GlcNAc transferase OGT. Thus, it cannot be simply interpreted as a readout of glucose availability or HBP activation in these tissue samples. The staining pattern detected by the CDT110.6 antibody was less marked but followed the same trend as RL2 staining. It is conceivable that in a heterogeneous bulk tumor tissue, the relative high abundance of *O*-GlcNacylated proteins (Trinidad et al, 2012) conceals CDT110.6 cross-reaction with the less abundant N-GlcNAc2–modified proteins occurring in distinct cell clusters.

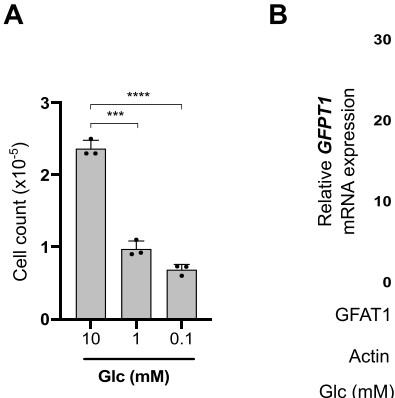

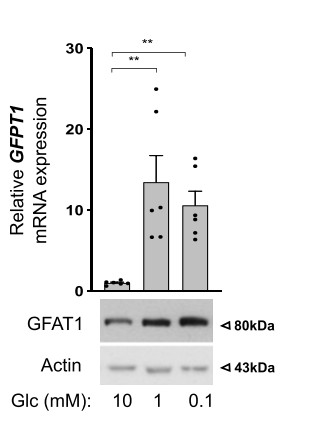

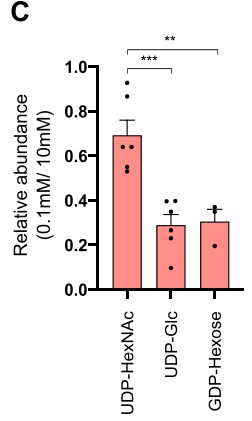

**Figure 2. UDP-HexNAc levels are preserved upon glucose shortage.**
**(A)** Cell counts from human bronchial epithelial cells (HBECs) cultured under sufficient (10 mM) or low (1 and 0.1 mM) glucose (Glc) conditions. ****$P < 0.0001$, ***$P < 0.001$ (unpaired $t$ test, n = 3 independent experiments, mean ± SEM). **(A, B)** Upper part: mRNA levels of GFAT1 in HBECs grown for 48 h as in (A). **$P < 0.01$ (unpaired $t$ test, n = 3 independent experiments, duplicate measurements, mean ± SEM). **(A)** Bottom part: representative Western blotting of GFAT1 in HBECs cultured as in (A). Actin is used as a loading control. **(C)** LC–MS/MS analyses of N-glycan nucleotide sugar levels in HBECs grown for 48 h under sufficient (10 mM) or low (0.1 mM) glucose conditions, presented as relative abundance normalized against 10 mM. ***$P < 0.001$, **$P < 0.01$ (unpaired $t$ test, n = 6 separate experiments, mean ± SEM).
Source data are available for this figure.

## UDP-GlcNAc production withstands impaired *N*-glycan elongation by mannose residues in the ER under glucose scarcity

Having shown that UDP-GlcNAc levels were not rate-limiting for *N*-glycan generation in the ER under low-glucose conditions, we hypothesized that the up-regulation of HBP's terminal metabolite upon glucose shortage should increase *N*-GlcNAc2–containing structures. This can be achieved by supplementing cells with exogenous GlcNAc that is taken up by pinocytosis and then largely converted to UDP-GlcNAc (Boehmelt et al, 2000; Wellen et al, 2010;

Abdel Rahman et al, 2013). GlcNAc supplementation to transformed HBECs cultured in 0.1 mM glucose substantially expanded UDP-HexNAc levels (Fig 4A), and this was accompanied by an increase in *O*-GlcNAcylation (Fig 4B, RL2 antibody). Paradoxically, GlcNAc supplementation diminished the immunodetection of some *N*-GlcNAc2–modified proteins (Fig 4B, CTD110.6 antibody), suggesting a loss of epitope recognition by the CTD110.6 antibody. These findings raised the intriguing possibility that maintaining UDP-GlcNAc production under low-glucose conditions may promote *N*-glycan elongation in the ER despite limited GDP-mannose availability.

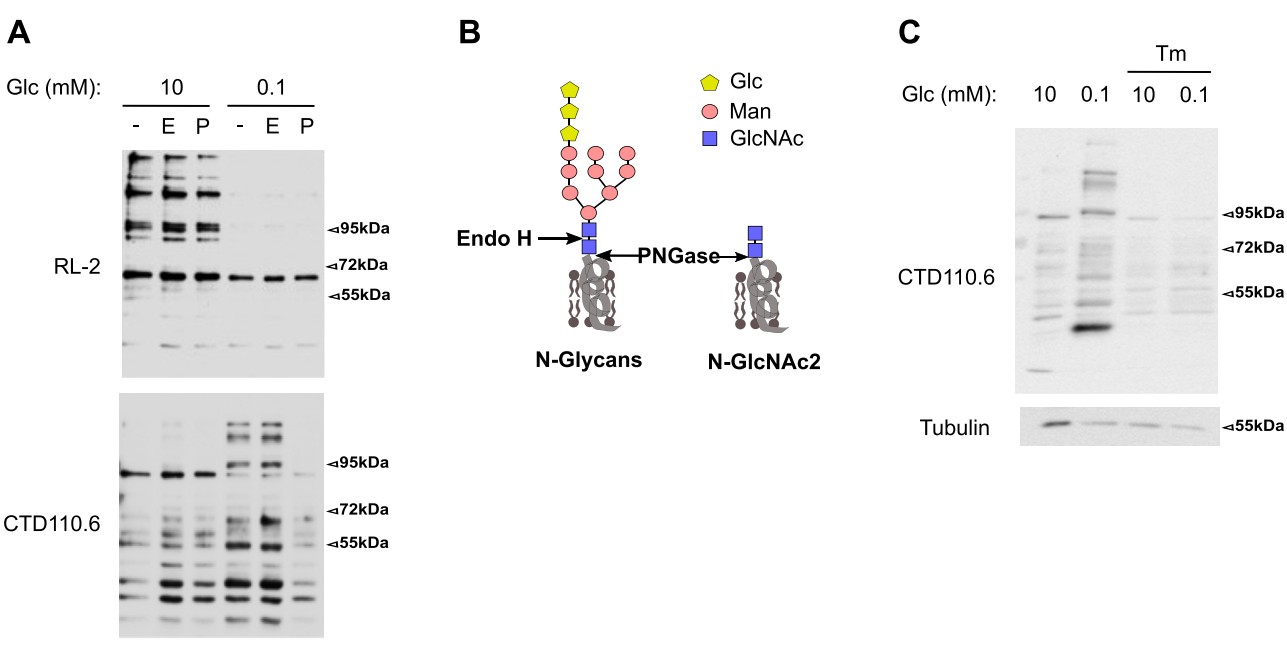

**Figure 3. Preserved levels of UDP-GlcNAc upon glucose shortage sustain *N*-glycosylation.**
**(A)** Representative Western blotting of *O*-GlcNAcylation modifications (RL2 antibody) and *N*-GlcNAc2–modified proteins (CTD110.6 antibody) in human bronchial epithelial cells cultured for 48 h in 10 or 0.1 mM glucose (Glc) conditions. Cellular lysates were treated with none (–), Endo H (E) or PNGase (P) enzymes. Tubulin is used as a loading control. **(B)** Schematic representation of core N-glycans and of *N*-GlcNAc2 structures (Glc, glucose; Man, mannose). Cleavage sites of PNGase and Endo H enzyme are indicated with arrows. **(A, C)** Representative Western blotting of *N*-GlcNAc2–modified proteins in human bronchial epithelial cells grown as in (A) and treated or not with tunicamycin (Tm). Tubulin is used as a loading control.
Source data are available for this figure.

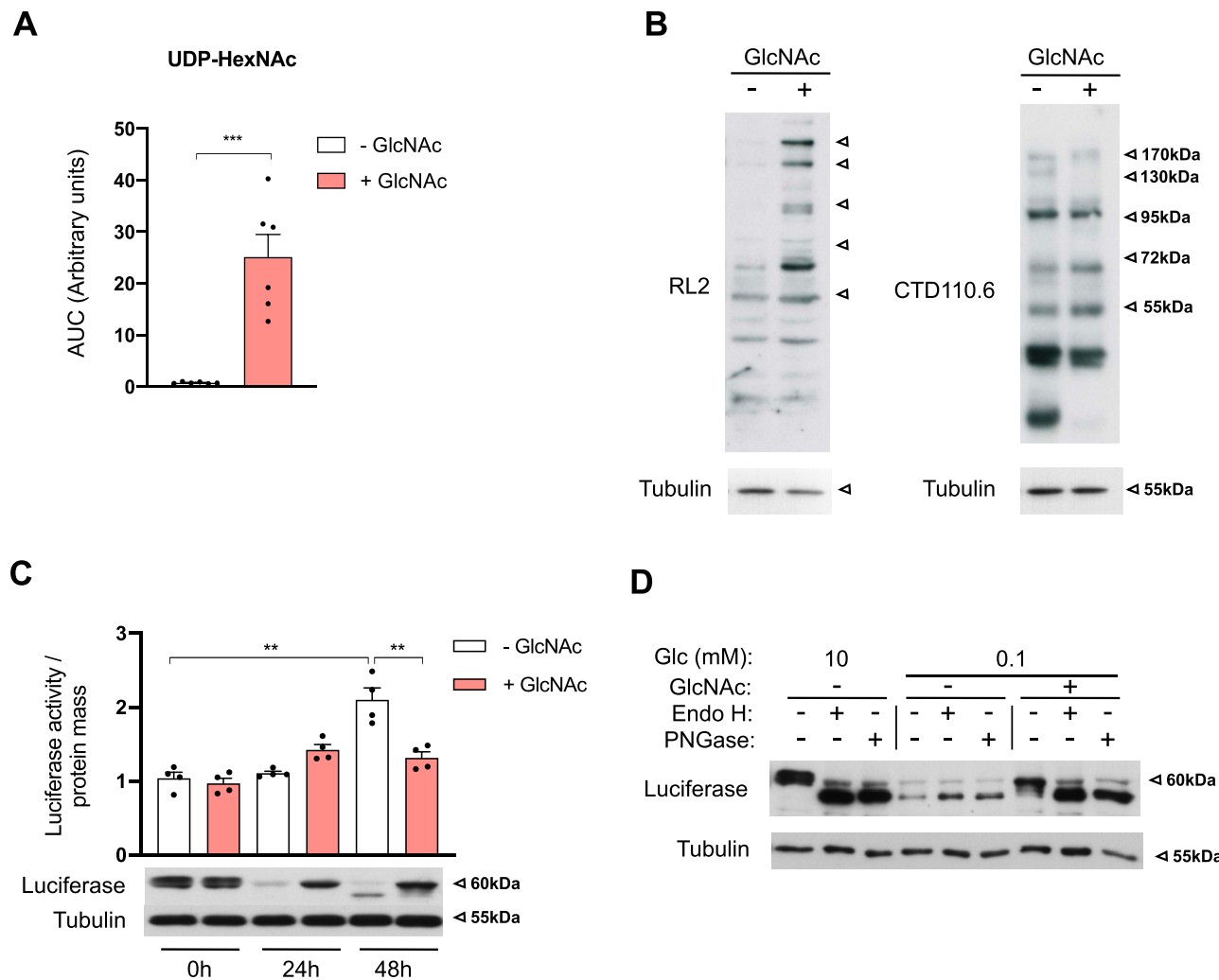

**Figure 4. Hexosamine biosynthetic pathway up-regulation relieves a low glucose-induced deficiency in *N*-glycan elongation in the ER.**
**(A)** Abundance of UDP-HexNAc in human bronchial epithelial cells cultured for 48 h in 0.1 mM glucose and supplemented or not with 25 mM GlcNAc, presented as the ratio of the area under the curve (AUC) of compound of interest/AUC of internal standard. ***$P < 0.001$ (unpaired $t$ test, n = 6 independent experiments, mean ± SEM). **(A, B)** Representative Western blotting of *N*-GlcNAc2–modified proteins (CTD110.6 antibody) and *O*-GlcNAcylation modifications (RL2 antibody) in human bronchial epithelial cells cultured as in (A). Tubulin is used as a loading control. **(C)** Upper part: luciferase assay of Hek293 ER-LucT cells grown for 24 and 48 h in glucose-limited conditions (0.1 mM) with or without GlcNAc supplementation. **$P < 0.01$, *$P < 0.05$ (unpaired $t$ test, n = 4 independent experiments, mean ± SEM). Bottom part: representative Western blotting of luciferase in Hek293 ER-LucT cultured under the same conditions. Tubulin is used as a loading control. **(D)** Representative Western blotting of luciferase in Hek293 ER-LucT grown for 48 h in 10, 1 or 0.1 mM glucose (Glc)-containing medium with or without GlcNAc supplementation. Cell lysates were treated or not with Endo H or PNGase. Tubulin is used as a loading control.
Source data are available for this figure.

To further examine this hypothesis, we used a luciferase reporter (ER-LucT) modified for ER translation, in which *N*-glycosylation interferes with its ability to display a bioluminescent signal (Contessa et al, 2010). We stably expressed this reporter in HEK293 cells, in which detection of proteins bearing *N*-linked GlcNAc2 structures in low-glucose conditions can be prevented by exogenous mannose supplementation (Sasaoka et al, 2018). Maintaining ER-LucT HEK293 cells in 0.1 mM glucose for 48 h led to an increased in luciferase activity and the appearance of a faster migrating band corresponding to nonglycosylated ER-LucT proteins (Fig 4C). GlcNAc supplementation concomitantly reduced the bioluminescent activity and the hypoglycosylation of ER-LucT, indicating that it is sufficient to restore glycosylation-mediated disruption of ER-LucT

activity. The reduced expression of ER-LucT upon low glucose was likely caused by low glucose-induced ER stress that attenuates global translation (Harding et al, 2000), which can be relieved by exogenous GlcNAc (Huber et al, 2013; Palorini et al, 2013; Song et al, 2018). Of note, the GlcNAc effect on glycosylation appeared independent of the protein translation rate (see below). Fig 4D shows that when cells were cultured in sufficient glucose (left hand) or low glucose supplemented with GlcNAc (right hand), both PNGase and Endo H treatment led to the immunodetection of lower molecular weight bands corresponding to nonglycosylated ER-LucT. Thus, GlcNAc supplementation promotes the elongation of *N*-linked structures with mannose residues as they are cleaved by Endo H glycosidase (Trimble & Tarentino, 1991; Freeze & Kranz, 2010).

These findings were unexpected given that exogenous GlcNAc is not thought to be catabolized and therefore to provide alternative sources of mannose residues. Nonetheless, they show that UDP-GlcNAc production under low glucose relieves a mannose-related deficiency in *N*-glycan elongation in the ER, the precise nature of which requires further investigation.

## UDP-GlcNAc production upon low glucose counterbalances the loss of surface expression for a subset of glycoproteins that includes EGFR

Although the above results indicated that UDP-GlcNAc production upon low glucose suffices to restore some degree of *N*-glycan elongation in the ER, they do not provide information on the consequences for endogenous proteins. Because *N*-glycosylation functions as a sorting signal for the transport of proteins to sub-cellular localizations (Roth, 2002), we profiled HBEC surface protein expression using biotinylation-based purification coupled with MS-based proteomic characterization. Preliminary results requiring experimental replication to be fully validated, suggested that glucose shortage for 48 h caused a decreased expression of proteins that were predicted to be localized on the cell membrane and that GlcNAc supplementation rescued, at least partially, the surface expression for a subset of these (Fig 5A and Table S2). At this stage of the study, we chose to validate the possible rescued surface expression of EGFR because (i) it was a top hit in our proteomic screening, (ii) it is highly *N*-glycosylated, and (iii) its transport to the plasma membrane and functions are sensitive to aberrant glycosylation (Sambrooks et al, 2018). FACS analysis confirmed that glucose shortage to 0.1 mM reduced EGFR surface expression (Fig 5B), which was not reflective of reduced gene expression (Fig S3A). GlcNAc supplementation did not affect the level of EGFR surface expression in the 10 mM glucose condition but efficiently restored it after glucose limitation. Thus, UDP-GlcNAc production upon low glucose is sufficient to relieve a default preventing cell surface expression of the EGFR.

Next, we examined whether, under low-glucose condition, the usage of the residual amount of UDP-GlcNAc prevented the reduction in EGFR surface expression. To test this, we used tunicamycin that impairs UDP-GlcNAc usage for N-glycan production (Keller et al, 1979). Cells were treated with 0.5 µg/ml of tunicamycin, a low dose found in preliminary experiments to limit its cytotoxic potential in HBECs. In cells cultured under sufficient glucose condition (10 mM), tunicamycin treatment for 24 h led to the appearance of an EGFR isoform with a smaller molecular weight of 130 kD (Fig 5C), consistent with an impaired *N*-linked glycosylation of the receptor as previously described (Contessa et al, 2008). Conversely, low glucose condition for 48 h induced the expression of EGFR isoforms with intermediate molecular weights of 145 kD instead of fully mature 175-kD EGFR, indicating an altered pattern of EGFR glycosylation. A role for glucose limitation in EGFR hypo-glycosylation, rather than a limitation in another nutrient, was further supported by competitively inhibiting the glucose meta-bolism with 2-deoxy-D-glucose (2-DG) upon sufficient glucose condition (Fig S3B). The low glucose-induced intermediate molecular weights of EGFR were replaced by the nonglycosylated 130-kD isoform upon 24 h of tunicamycin treatment. Thus, the residual

amount of UDP-GlcNAc is used to fuel some degree of EGFR *N*-glycosylation. Concurrently, the reduction in cell surface EGFR detected upon low glucose was further aggravated by tunicamycin treatment (Fig 5D), strongly suggesting that altered EGFR glycosylation upon low glucose contributed to its surface expression. Hence, under low glucose availability, the residual amount of UDP-GlcNAc fueling the *N*-glycosylation pathway partly compensates for the reduced expression of EGFR at the cell surface of HBECs.

## UDP-GlcNAc production under low glucose promotes EGF stimulation of anchorage-independent growth and is correlated with EGFR activation in LUAD tissues

Next, we wondered whether exogenous GlcNAc supplementation in low glucose could reinstate normal EGFR biological functions. As for the ER-LucT construct (see above), GlcNAc supplementation prevented the low glucose–induced hypoglycosylation of EGFR (Fig 6A). It is of note that GlcNAc supplementation also partly counteracted 2-DG–reduced EGFR glycosylation (Fig S3B). Although EGFR amounts were reduced by glucose shortage, prevention of protein hypo-glycosylation by exogenous GlcNAc was not linked to protein translation rate in the ER. Indeed, Fig S4A shows that GlcNAc counterbalanced *N*-glycosylation loss of the transmembrane protein PD-L1 whose translation in the ER is not affected by ER tress (Chou et al, 2020). Predictably, the loss of EGFR surface expression resulted in reduced EGF-induced phosphorylation of EGFR and of its direct substrates Shc, Gab1, and PLCg (Fig 6A). In contrast, GlcNAc-restored surface EGFR was effectively activated by EGF as shown by the phosphorylation of EGFR and of its downstream targets. Similar results were recapitulated in the H358 cell line (Fig S4B). Because EGFR is one of the most mutated oncogenes in LUAD (Skoulidis & Heymach, 2019), we wondered whether EGFR mutants signaling could also be affected by GlcNAc supplementation upon low glucose. Although exogenous GlcNAc prevented low glucose–induced hypoglycosylation of mutants EGFR, it did not significantly affect their autophosphorylation or the phosphorylation of their downstream targets (Fig S4C). Thus, the stimulation of the UDP-GlcNAc level upon low-glucose conditions is sufficient to rescue wild-type EGFR signaling.

Then, we examined whether UDP-GlcNAc–rescued EGFR signaling translated into cellular protective functions in low glucose conditions. Transformed HBECs are dependent on EGF for cell proliferation in traditional 2D monolayer culture conditions (Sato et al, 2013). However, in a low-glucose condition, GlcNAc-restored EGFR signaling did not increase cell growth in standard cell cultures (Fig S4D). We thus grew these cells in a 3D system, in which cells form spheroids when cultured on a nonadherent substrate and that is thought to better recapitulate the in vivo tumor microenvironment (Langhans, 2018). Using ATP content as a surrogate marker of spheroid cell number/viability (Zanoni et al, 2016), we found that EGF and/or GlcNAc supplementation did not affect spheroid growth in the 10-mM glucose condition (Fig S4E). However, the 0.1-mM glucose condition resulted in a reduction in spheroid size (Fig 6B) and a similar trend in their ATP content (Fig 6C). In striking contrast to the effects in the 2D monolayer system, supplementation with GlcNAc and EGF of the 0.1-mM glucose condition increased the size and viability of spheroids (Fig 6B and C). These

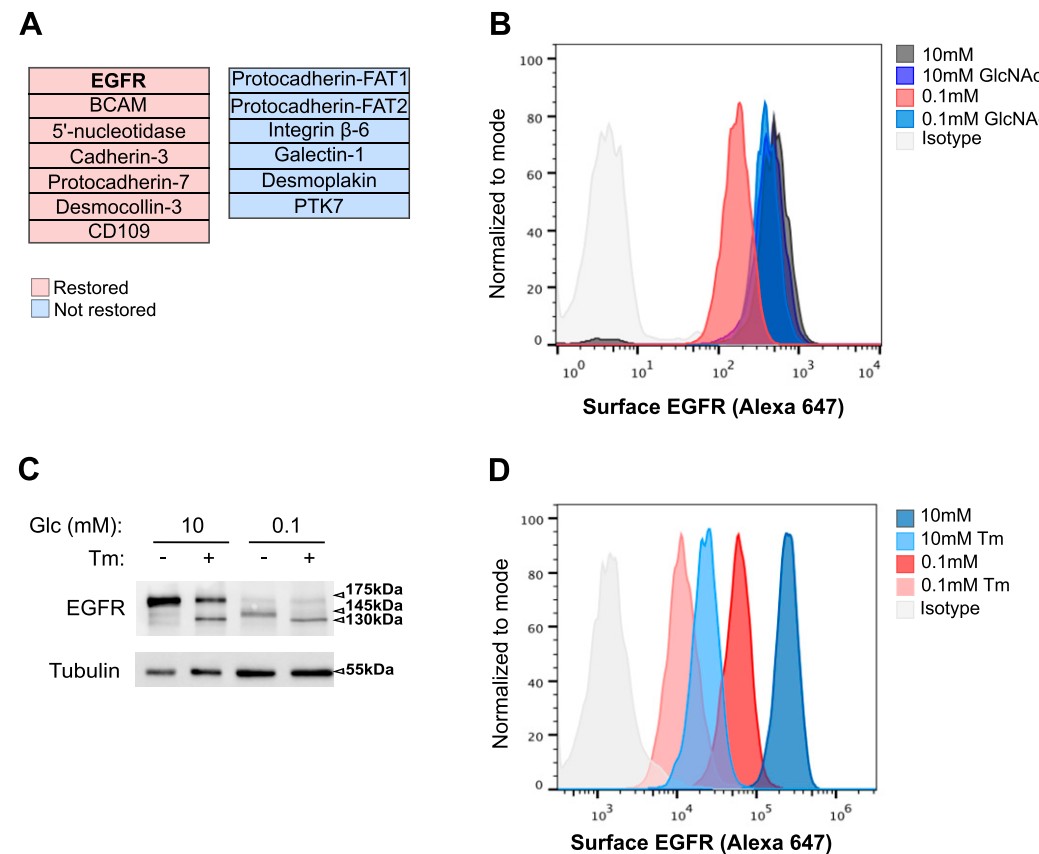

**Figure 5. Hexosamine biosynthetic pathway activation upon low glucose counterbalances the loss of surface epidermal growth factor receptor (EGFR) expression.**
**(A)** Cellular surface proteome analysis of human bronchial epithelial cells (HBECs) cultured for 48 h in 10 or 0.1 mM glucose (Glc), with and without GlcNAc supplementation. Shown are proteins manually curated for being localized on the cell membrane and whose expression was down-regulated in the 0.1 mM glucose condition and restored or not by GlcNAc supplementation. **(A, B)** Representative FACS analysis of EGFR surface expression in HBECs grown as in (A). **(C)** Representative Western blotting of EGFR in HBECs grown for 48 h in 10 or 0.1 mM glucose conditions and treated or not with tunicamycin (Tm) for the last 24 h of culture. Tubulin is used as a loading control. **(C, D)** Representative FACS analysis of EGFR surface expression in HBECs grown and treated as in (C).
Source data are available for this figure.

results were reproduced using the H358 cell line (Fig S4F). At the molecular level, GlcNAc supplementation restored EGFR signaling (Fig 6D) and stimulated the expression of the malignancy-associated transcription factor *SOX2* (Chou et al, 2013; Novak et al, 2020) (Fig 6E), used here as a readout of gene transcription efficiency in the 3D system under low-glucose condition. Of note, this EGF-induced *SOX2* was not observed in the 2D culture system (Fig S4G). Together, these results indicate that upon glucose scarcity, the production of UDP-GlcNAc per se can rescue a loss of EGFR signaling to promote anchorage-independent growth.

Next, we investigated a relationship between HBP stimulation and EGFR activation in patient samples, using immunohistochemical (IHC) analysis of a LUAD tissue microarray (TMA; n = 145) (Serra et al, 2018). GFAT1 staining was predominant in tumor cells (Fig 7A). Few samples had a negative or low H-score, leading to two distinct groups with medium (n = 42) or high (n = 96) GFAT1 staining. EGFR phosphorylation, reflecting its activation, occurs in ~45% of LUAD samples (Kanematsu et al, 2003). Consistently, ~46% of the samples were positive for p-EGFR (Fig 7B). Clinical annotation of the TMA showed that ~12% of the samples were mutated for EGFR (Serra

et al, 2018), indicating that wild-type EGFR accounted for most of positive samples. Notably, p-EGFR–positive samples were more abundant in the group with high expression of the HBP rate-limiting enzyme GFAT1 (Fig 7C). Thus, these findings support a positive association between HBP stimulation and EGFR activation in LUAD tissues.

**COPII is up-regulated in LUAD and opposes the loss of EGFR cell surface expression in low glucose**

Intriguingly, Gene Ontology (GO) analysis of genes exhibiting the highest correlation with *GFPT1* expression in LUAD TCGA datasets indicated an enrichment in terms related to ER-Golgi vesicle transport (Fig S5A and B), which is mediated by the coat protein complex (COP) I and II (Brandizzi & Barlowe, 2013). We therefore assessed the expression of COP genes in LUAD and LUSC. COPI component transcripts were increased in tumor tissues with no marked difference between LUAD and LUSC (Fig 8A). In contrast, the expression of COPII genes was distinctly up-regulated in LUAD. Consistently, immunoblotting of the COPII component SEC24D protein using the same samples as in Fig 1C showed that it was more

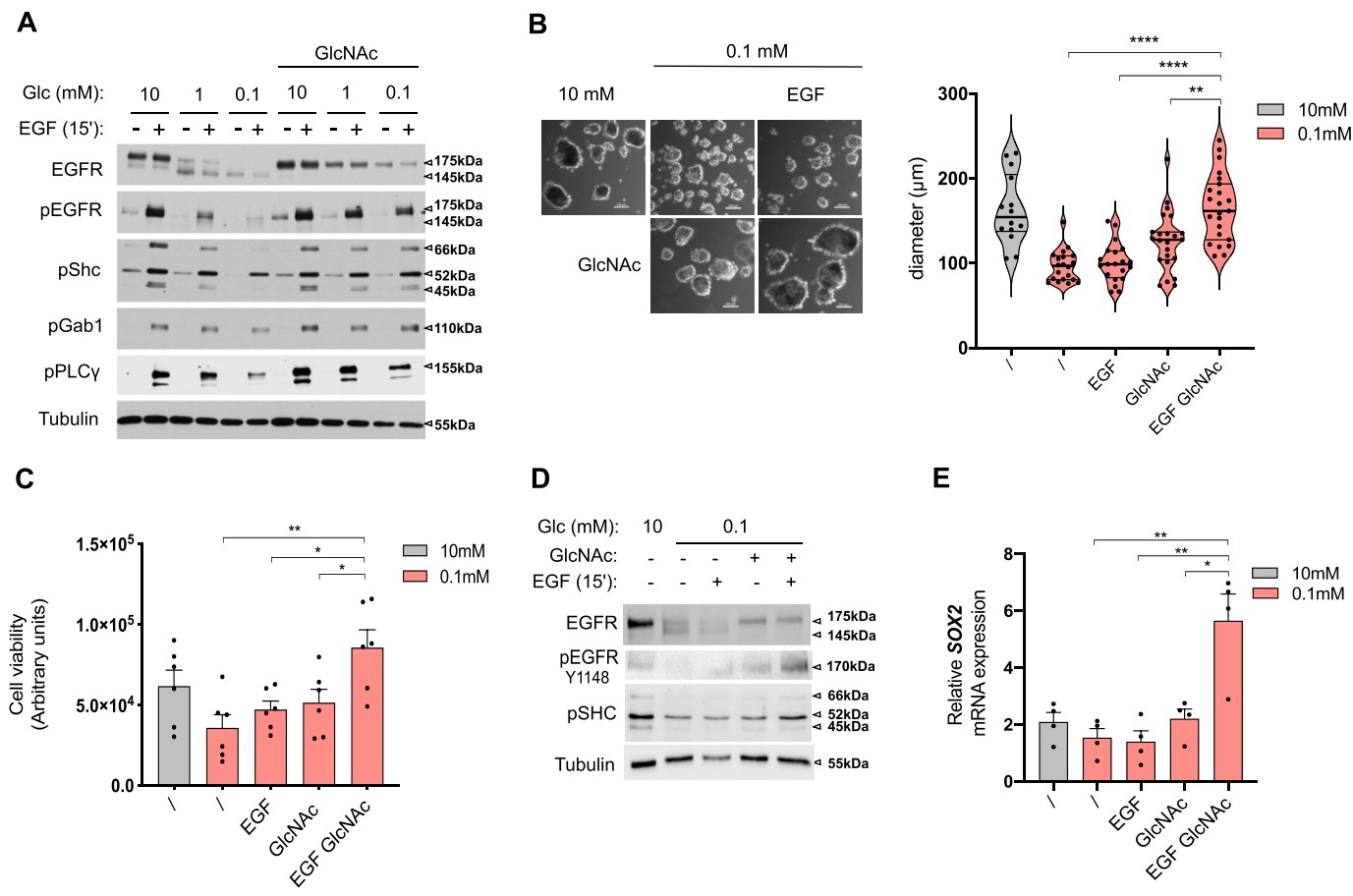

**Figure 6. Hexosamine biosynthetic pathway activation upon glucose scarcity promotes EGF stimulation of anchorage-independent growth.**
**(A)** Representative Western blotting of the indicated proteins in human bronchial epithelial cells (HBECs) cells grown for 48 h in 10, 1, or 0.1 mM glucose (Glc)-containing medium with and without GlcNAc and/or EGF supplementation. Tubulin is used as a loading control. **(B)** Representative images (right hand) and size analysis (left hand) of HBEC spheroids grown under 10 mM glucose control condition or 0.1 mM glucose without or with GlcNAc and/or EGF supplementation. ****$P < 0.0001$, **$P < 0.01$ (unpaired $t$ test, mean ± SEM). Scale bars = 100 $\mu$m. **(B, C)** Cell viability assay of HBECs spheroids grown and treated as in (B). **$P < 0.01$, *$P < 0.05$ (unpaired $t$ test, n = 6 independent experiments, mean ± SEM). **(B, D)** Representative Western blotting of the indicated proteins in HBEC spheroids grown and treated as in (B). **(E)** SOX2 mRNA level in HBEC spheroids grown and treated as in (B). **$P < 0.01$, *$P < 0.05$ (unpaired $t$ test, n = 4 independent experiments, mean ± SEM).
Source data are available for this figure.

abundant in LUAD than in LUSC tissues (Fig 8B). These results indicate that HBP up-regulation in LUAD is accompanied by an increase in COPII components.

Notably, it has been shown that EGFR expression can be regulated by the COPII components SEC23B and SEC24D (Scharaw et al, 2016). We found that low glucose increased the mRNA expression of COPII components in transformed HBECs, among which *SEC24D* was strongly up-regulated (Fig 8C). Accordingly, Western blot analysis of the same samples as in Fig 2B showed that the protein level of SEC24D was also increased (Fig 8D). Thus, we addressed the outcome of COPII up-regulation on EGFR surface expression under low-glucose condition. Silencing *SEC24D* expression using siRNAs did not affect EGFR surface expression under sufficient glucose condition (Fig 8E). In contrast, *SEC24D* silencing further diminished the remaining cell surface EGFR detected in the low glucose condition. A similar trend was observed when using H358 (Fig S5C). Altogether, these data indicate that in addition to HBP's role, the up-regulation of SEC24D COPII coat component

upon low glucose availability also partly compensate for the reduced surface expression of EGFR.

# Discussion

Here, we show that the HBP and COPII up-regulations in LUAD counterbalance a loss of EGF-dependent cellular viability when glucose becomes scarce.

We found that the HBP is engaged to a higher extent in the LUAD than in LUSC subgroups, as indicated by *GFPT1* expression and metabolite production. These results substantiate previous findings of a differential expression of *GFPT1* in LUAD and LUSC (Zhang et al, 2018). Furthermore, LUAD can be distinguished from LUSC by a selective up-regulation of COPII-dependent vesicular transport that was correlated with *GFPT1* expression. Thus, these findings expand the sharp distinction between these two NSCLC subtypes at the molecular, pathological, and clinical levels (Relli et al, 2019).

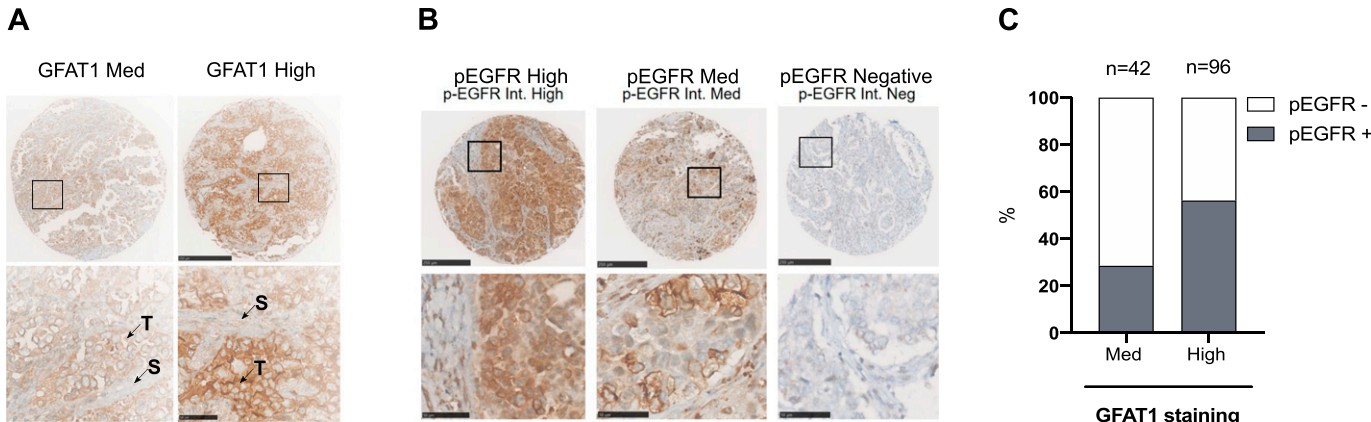

**Figure 7. High GFPT1 expression associates with increased phosphorylation of wild-type epidermal growth factor receptor (EGFR) in LUAD patient samples.**
**(A)** Representatives images of high and medium (Med) GFAT1 IHC of LUAD TMA sections. Scale bars = 250 μm. Bottom: enlarged view of insets. Scale bars = 50 μm. T, tumor cells; S, stromal cells. **(B)** Representatives images of high (n = 17), medium (n = 50), and negative (n = 78) staining of phosphorylated EGFR (p-EGFR) IHC of LUAD TMA sections. Scale bars = 250 μm. Bottom pictures are enlarged views from insets. Scale bars = 50 μm. **(C)** Percentage of samples positive (high and Med) or negative for p-EGFR within LUAD TMA groups displaying medium or high GFAT1 staining.

Overall glucose levels are limited in the lung tumor microenvironment compared with normal tissue (Urasaki et al, 2012; Wikoff et al, 2015; Hensley et al, 2016). This limitation conceals regional differences, comprising areas of hypoxia and nutrient scarcity because of insufficient vascular network and an imbalance of supply and demand (Garcia-Canaveras et al, 2019). The understanding of adaptive responses in cancer cells to these specific local nutritional conditions, and the resulting metabolic flexibility for cancer progression, is needed to unravel novel levers for therapy. In this context, our findings unveil that HBP's terminal metabolite maintenance and COPII up-regulation are part of the adaptive responses to low glucose condition in LUAD cell lines, which promote an EGFR-dependent cellular protective function. This protective function for wild-type EGFR might not be maintained in EGFR carrying oncogenic activating mutations because our findings suggested that EGFR mutant signaling was not significantly impacted by the maintenance of UDP-GlcNAc levels upon low glucose. A possible reason might be that in contrast to wild-type EGFR, mutant EGFR signaling is constitutively activated and does not depend on the binding of an activating ligand at the cell surface (Pines et al, 2010). The EGFR-dependent protective function was not spotted in traditional 2D culture but played a predominant role when cells were cultured in 3D conditions. Notably, cells grown in 3D appear to better recapitulate the physiology of the complex microenvironment occurring in tumors (Langhans, 2018) and model more accurately vulnerabilities in oncogenic signaling pathways (Biancur et al, 2021). Thus, our data argue in favor of systematically using 3D culture to unmask mechanisms of cellular adaptation to nutrient shortage.

It has been reported that COPII proteins control EGFR vesicular transport from the ER upon EGF stimulation (Scharaw et al, 2016). Here, we show that the SEC24D COPII component is also engaged in sustaining surface EGFR expression when glucose becomes scarce. This regulation by COPII was dispensable under a sufficient glucose condition, pointing to the existence of stress-specific functions for COPII. A large number of integral membrane and secreted proteins are transported via the COPII pathway, and it is therefore tempting to speculate that COPII up-regulation may contribute to different aspects of malignant cell adaptation to low glucose.

Glucose availability regulates the levels of the three nucleotide sugars (UDP-GlcNAc, GDP-mannose, and UDP-glucose) serving as sugar donor substrates for DLOs synthesis, the *N*-glycan precursors (Harada et al, 2013). Nevertheless, consistent with previous studies on DLO biosynthesis in glucose-deprived cells (Nakajima et al, 2010; Harada et al, 2013), we highlighted that the amount of the nucleotide pool containing UDP-GlcNAc was the less affected by glucose scarcity. Better preserved UDP-GlcNAc amounts may result from the increased channeling of traces of glucose-derived metabolites into the HBP via the up-regulated rate-limiting enzyme GFPT1 and/ or from GlcNAc salvage from recycled glycoconjugates via the N-acetylglucosamine kinase (NAGK) (Boehmelt et al, 2000; Wellen et al, 2010). Interestingly, a role for the GlcNAc salvage pathway in supporting pancreatic ductal adenocarcinoma tumor growth has been recently described (Campbell et al, 2021). Although we found that NAGK expression is not regulated by low glucose in transformed HBECs (Fig S6), NAGK activity might also be regulated through post-translational modifications (Campbell et al, 2021), but very little is currently known about such a regulation. In addition, we found that the persisting UDP-GlcNAc pool was actively used to fuel the *N*-glycosylation pathway, whereas the *O*-GlcNAcylation pathway was strongly down-regulated, advocating for the need of *N*-glycosylation pathway robustness upon nutrient perturbations. However, usage of UDP-GlcNAc in the face of limiting GDP-mannose leads to incompletely assembled DLOs that are rapidly degraded (Harada et al, 2013) and thus are rarely transferred to proteins in cells (Harada et al, 2021). Yet, the detection by others and us of low glucose-induced protein-linked GlcNAc2 structures (so called chitobiose) reveals that under harsh conditions, an incomplete *N*-glycan precursor is transferred into the lumen of the ER and subsequently added to proteins. In the case of mannosylation deficiency-associated congenital disorders of glycosylation (CDG), the chitobiose occurrence enables the production of

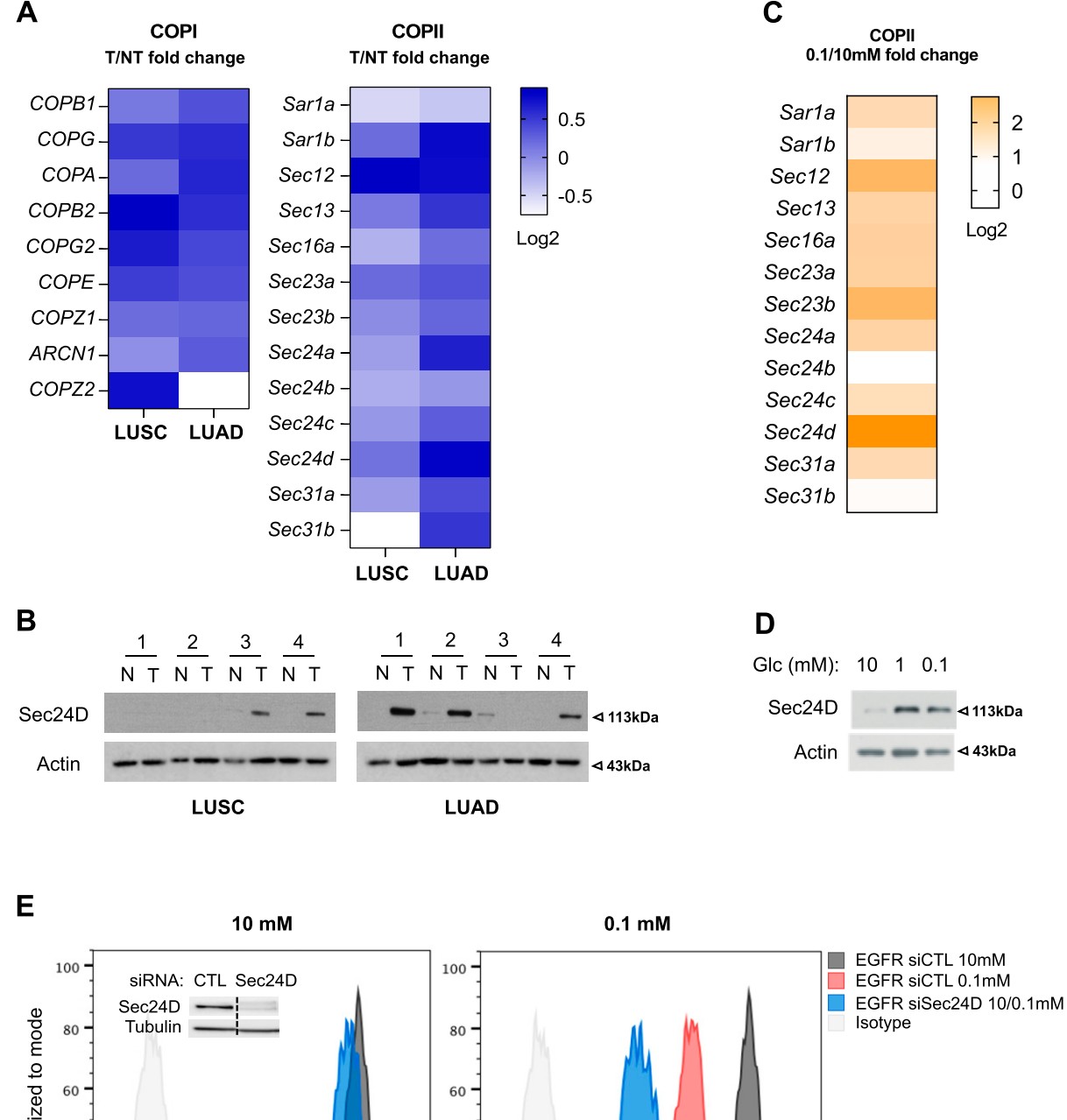

**Figure 8. COPII up-regulation counterbalances the loss of epidermal growth factor receptor cell surface expression in low glucose condition.**
**(A)** The Cancer Genome Atlas analysis of mRNA levels of COPI and COPII components in LUAD and LUSC, presented as Log$_2$ fold changes in expression between tumor (T) and nontumor (NT) tissue. **(B)** Western blotting of SEC24D in LUAD and LUSC paired nontumor (N) and tumor (T) tissue samples (same membranes as the one used for Fig 1C). Actin is used as a loading control. **(C)** mRNA level of COPII components grown for 48 h under sufficient or limited glucose conditions and presented as log$_2$ fold changes in expression between 0.1 and 10 mM of glucose. **(D)** Representative Western blotting of Sec24D in human bronchial epithelial cells cultured in sufficient (10 mM) or low (1 and 0.1 mM) glucose (Glc) conditions (same membrane as the one used for Fig 2B). Actin is used as a loading control. **(E)** Representative FACS analysis of epidermal

nonconventional N-glycans at the cellular surface (Ng et al, 2016; Zhang et al, 2016). Here, we show that such occurrence upon low glucose allowed the rescue of *N*-glycan elongation by mannose residues, unlocking protein transport to the cell surface. Intriguingly, this safeguarding mechanism might regulate glycosylation for a subset of membrane proteins only, which reflect the recently described translocon-associated regulation of glycosylation for a subset of ER-translated protein clients (Phoomak et al, 2021). Hence, starvation or starvation-like stresses are associated with some leakage in the quality control processes of *N*-glycosylation that confers adaptive ways for cells to maintain, possibly in a selective manner, a critical post-translational modification.

Previously, it has been shown that uptake of exogenous GlcNAc by glucose-starved cells leads to the continuation of UDP-GlcNAc biosynthesis and to an increased branching of the N-linked carbohydrate chains in the Golgi bodies (Wellen et al, 2010). Here, we unveiled that the maintenance of HBP's terminal metabolite is also an important actor for the rescue of *N*-glycosylation in the ER apparatus. In agreement with this, GlcNAc supplementation of glucose-deprived cells was shown to promote as effectively as glucose, the N-glycosylation of SREBP-cleavage activating protein in the ER (Cheng et al, 2015). How GlcNAc supplementation relieves the impaired mannose addition during *N*-glycan building is currently unclear. Nevertheless, these findings unravel an unforeseen function of HBP's terminal metabolite in maintaining the *N*-glycosylation of proteins in the ER to help cells adapt to low glucose (Fig 9).

In conclusion, this work reveals that LUAD distinctive features provide protective functions to the malignant cells facing low glucose stress, strengthening the therapeutic interest in targeting the HBP for this disease. In addition, it raises questions for future investigations on the quality control process of *N*-glycosylation in the ER under stress.

# Materials and Methods

### Cell culture, reagents, and treatments

Transformed HBECs (donated by Dr. JD Minna) were grown in Keratinocyte Serum-Free Medium supplemented with 50 µg/ml bovine pituitary extract (Gibco), 5 ng/ml EGF (Peprotech), and 1% vol/vol penicillin/streptomycin solution (Gibco) as described (Sato et al, 2013). NCI-H358, PC9, and Hek293-LucT cells were grown in RPMI 1640 supplemented with 10% FBS and 1% vol/vol penicillin/streptomycin solution. Cells were maintained at 37°C with 5% $CO_2$ throughout the experiments. 2-DG was from Sigma-Aldrich. For glucose shortage experiments, cells were rinsed with phosphate-buffered saline and incubated in DMEM medium containing the indicated amount of glucose (Gibco) for 48 h, and the medium was refreshed at 24 h. When indicated, DMEM medium was supplemented with 25 mM N-acetylglucosamine

(GlcNAc) (Sigma-Aldrich), 5 ng/ml EGF, or 0.5 µg/ml tunicamycin (Sigma-Aldrich). Three-dimensional spheroids were generated by plating HBECs or NCI-H358 cells passed through a 40-µm cell strainer (Corning), into ultra-low attachment plates or flasks (Corning) and maintained in DMEM medium containing the indicated amount of glucose for 5 d. To examine carbohydrate structure, cellular lysates were treated with PNGase F (P0704S; BioLabs) or Endo H glycosidase (P0702S; BioLabs) according to manufacturer's instructions.

The silencing of SEC24D was achieved using a combination of small interfering RNA, designed and synthesized by Eurogentec: siSec24D#1 (5'-GGA-GAA-GUC-UUU-GUU-CCU-U55-3' and 3'-AAG-GAA-CAA-AGA-CUU-CUC-C55-5') and siSec24D#2 (5'-CUG-UCU-UAC-CCA-GGA-GGC-U55-3' and 3'-AGC-CUC-CUG-GGU-AAG-ACA-G55-5'). Briefly, cells were transfected in six-well plates (typically 400,000 cells per well) using a combination of 75 nM of siRNA and 12 µl of HiPerFect reagent (QIAGEN) according to the manufacturer's reverse transfection protocol. Thirty hours post-transfection, cells were further submitted or not to glucose shortage as indicated.

### TCGA NSCLC cohort and gene set enrichment analyses

Publicly accessible TCGA NSCLC RNA-seq data on lung adenocarcinoma (n = 514) and lung squamous cell carcinoma (n = 502), or solid tissue normal/primary tumor paired data of lung adenocarcinoma (n = 58) and lung squamous cell carcinoma (n = 52), were used to analyze gene expression presented as RSEM ($Log_2$)-normalized count from the Broad Institute (https://gdac.broadinstitute.org/).

A list of 38 best-correlated genes to GFPT1 in TCGA lung adenocarcinoma datasets (Pearson's R > 3.29; *P* < 0.001) was identified using R (version 3.6.2) (http://www.r-project.org). Pathway enrichment using gene set enrichment analysis was conducted using Molecular Signatures Database (MsigDB version 7.4) and the subset of GO, GO Biological Process ontology (GO: BP) (False Discovery Rate q-value < 0.05).

### Tissue microarray cohort, immunostaining, and scoring

A lung adenocarcinoma tissue microarray (TMA) was constructed from 152 patient biopsies by the Hospice Civils de Lyon (CRB Tumorothèque) (Serra et al, 2018). Seven samples were not considered herein as they were either duplicates or metastases. TMA slides were deparaffinized and dehydrated. To block endogenous peroxidases, tissue sections were incubated in 5% hydrogen peroxide solution. Staining was performed using an automated immunostainer (Ventana Discovery XT; Roche), with antibodies raised against p-EGFR Tyr1068 (2234S; Cell Signalling) or GFPT1 (Ab125069) according to manufacturer's instructions. TMA staining intensity was assessed by pathologist visual scoring as follows: 0 (negative), 0–50 (low), 50–200 (medium), and 200–300 (high) H-score.

---

growth factor receptor surface expression in human bronchial epithelial cells grown for 48 h in medium containing 10 or 0.1 mM glucose and silenced or not for Sec2D. Inset shows Sec2D protein amounts upon silencing with siRNAs, and tubulin is used as a loading control.
Source data are available for this figure.

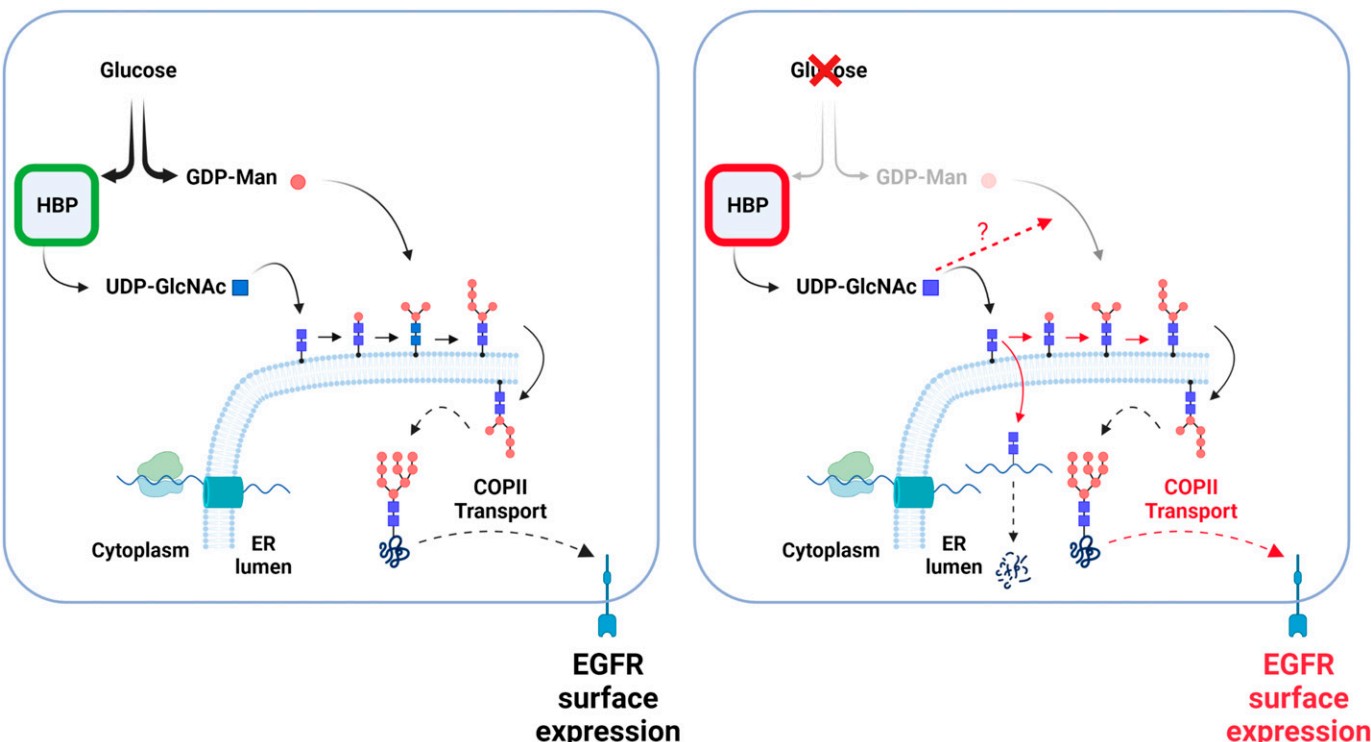

**Figure 9. Illustration of *N*-glycan building rescue by UDP-GlcNAc under glucose scarcity.**
(Left hand) In glucose sufficient condition, substrates are provided to produce UDP-GlcNAc and GDP-Man, two nucleotide sugars necessary for the building of *N*-glycans in the ER. Building of *N*-glycans and their transfer onto newly synthesized proteins (wavy line), including epidermal growth factor receptor, enhances proper protein folding allowing their transport via COPII vesicle to Golgi and plasma membrane. (Right hand) Glucose shortage reduces GDP-Man levels, whereas those of hexosamine biosynthetic pathway's terminal metabolite are better maintained. This persisting UDP-GlcNAc pool is actively used to initiate *N*-glycan building, leading to incomplete *N*-glycan precursors that are nonetheless transferred onto protein. Proteins bearing only GlcNAc2 structures (chitobiose) are likely degraded (Isono et al, 2013). However, maintained UDP-GlcNAc production also promotes chitobiose elongation by mannoses, through yet unclear mechanisms, rescuing some protein folding. This adaptive mechanism (marked in red), together with a selective increase in COPII transport, unlocks protein transport to the cell surface and promotes an epidermal growth factor receptor-dependent cellular protective function.

## Cell extracts and Western blot analyses

Cultured cells were lysed in RIPA protein buffer containing proteases and phosphatases inhibitors (Roche). Protein concentration was determined using DC Protein Assay (Bio-Rad). Equal amounts of protein (15 µg) were separated by SDS–PAGE and then transferred onto nitrocellulose membranes (Bio-Rad). Membranes were incubated in blocking buffer, 5% milk, or BSA in Tris-buffered saline/Tween 20 (TBST), for 1 h at room temperature and then incubated overnight at 4°C with the appropriate primary antibody, diluted in TBST containing 5% milk or BSA. Membranes were washed three times with TBST, incubated for 1 h at room temperature with the appropriate secondary antibody in TBST containing 5% milk or BSA, and again washed three times with TBST. Detection was performed using the Clarity Western ECL Substrate (Bio-Rad) or SuperSignal West Femto Maximum Sensitivity Substrate (Thermo Fisher Scientific) for phoshoprotein antibodies. Representative Western blot of three independent experiments was shown. Scanned images of Western blot were quantified with ImageJ v.1.53 (Rueden et al, 2017).

Primary antibodies were purchased from Santa Cruz Biotechnology: CTD110.6 (sc-59623; 1:5,000) and RL2 (sc-59624; 1:500); from Cell Signalling Technology: EGFR (4267; 1:1,000), p-EGFR Tyr1068 (2234S; 1:1,000), p-EGFR Tyr1148 (4404; 1:1,000), p-Shc Tyr239/240 (2434; 1:1,000), p-Gab1 Tyr627 (3233; 1:1,000), p-PLCg1 Tyr783 (2821; 1:1,000), p-ERK (4370; 1:1,000), β-Tubulin (2146; 1:1,000), actin (8432; 1/1,000), PD-L1 (13684; 1:1,000), and HRP-linked anti-rabbit, and anti-mouse IgG secondary antibodies (7074, 7076; 1:10,000); from Abcam: GFPT1 (ab125069; 1:1,000), NAGK (ab203900; 1:1,000), and Sec24D (ab191566; 1:1,000); from Merck Millipore: Luciferase (2433432; 1:1,000); and from Proteintech: OGT (11576-2-AP; 1:1,000).

## RNA extraction and RT-qPCR

RNA was extracted using TRIzol (Invitrogen), and cDNA was synthesized using Superscript II reverse transcriptase (Invitrogen) with random primers (Invitrogen), according to the manufacturer's instructions. Quantitative PCR was performed on the CFX Connect Real-Time PCR System (Bio-Rad) using primers listed in the Supplementary material (Table S3) and SYBR Green Master Mix (Bio-Rad). Expression of target genes was normalized against endogenous RPS11 or HPRT mRNA levels. HPRT amounts proved to be more stable in 3D cell culture.

## Cell viability and SRB assays

Cell viability was determined using CellTiter-Glo 3D assay (Promega) according to manufacturer's instructions. Luminescence signal

(integration time 1 s) was measured using a microplate reader (Tecan, Infinite M200 PRO; Life Science). Cell viability is expressed as the mean percentage compared with the control.

Cellular mass upon glucose shortage was determined using the sulforhodamine B (SRB) assay. Cells were fixed with ice cold 10% trichloroacetic acid (TCA) for 1 h at 4°C. TCA was then removed and cells were air dried before adding 0.057% SRB solution. After 30 min of incubation, cells were rinsed with 1% acetic acid and air dried. The cell-incorporated dye was solubilized by adding a Tris base solution (pH 10.5), and the absorbance was measured at 510 nm in a microplate reader.

### Cell metabolite profiling by liquid chromatography–mass spectrometry (LC–MS/MS)

Metabolites were extracted using cold methanol/water mixture (70/30, vol/vol) from cultured cells washed and scrapped off in cold PBS or from 6 LUAD and 6 LUSC paired tissue samples homogenized cryogenically (obtained from the Biobank BB-0033-00025, University Côte d'Azur). All patients signed an informed consent to participate in the study. Supernatants were frozen using liquid nitrogen and sent for LC/MS–MS analysis (HCL; Lyon-Sud). Precipitated proteins were solubilized in RIPA containing 8 M urea and quantified using the Lowry method. A volume of lysate corresponding to 0.5–0.6 µg of proteins was used for each sample, and the labeled internal standard (CTP$_{13C}$) solution was added. UDP-HexNAc, GlcNAc-6-phosphate, UDP-glucose, and GDP-hexose were quantified by online extraction coupled with LC–MS/MS, as previously described (Machon et al, 2014). Briefly, online extraction is performed on a Oasis WAX column (Waters) and analytical separation on a Hypercarb column (Thermo Fisher Scientific). The mass spectrometer, a TSQ Quantum Ultra (Thermo Fisher Scientific), was operated in positive ion multiple-reaction monitoring mode for UDP-HexNAc and GlcNAc-6-phosphate. The following transitions were used: m/z 608→204, m/z 302→284, m/z 229→97 and m/z 349→137 for UDP-HexNAc, GlcNAc-6-phosphate, UDP-glucose, and GDP-hexose, respectively. Results were expressed as the ratio of the area under the curve of compound of interest/area of internal standard.

### Glucose and lactate quantification

Glucose and lactate concentrations were measured in cell culture supernatants collected at the indicated times, using the ARCHITECT C16000 automated analyzer (Abbott Laboratories).

### MS-based characterization of surface proteomes

Subconfluent HBEC-RL53 cells were rinsed twice with PBS supplemented with 0.1 mM CaCl2 and 1 mM MgCl2 (PBS+) and incubated with 1 mg/ml EZ-Link Sulfo-NHS-SS-Biotin (Thermo Fisher Scientific) in PBS+ for 30 min on ice. After removal of the supernatant, the residual biotinylation reagent was quenched with 100 mM glycine in PBS+, and the cells were harvested and lysed with Ripa buffer. Biotinylated cell surface proteins were purified using NeutrAvidin beads according to the manufacturer's instructions (Thermo Fisher Scientific), solubilized in Laemmli buffer and stacked on top of a

4–12% NuPAGE gel (Invitrogen). After staining with R-250 Coomassie Blue (Bio-Rad), proteins were digested in-gel using trypsin (modified, sequencing purity; Promega), as previously described (Casabona et al, 2013). The resulting peptides were analyzed by online nanoliquid chromatography coupled to MS/MS (Ultimate 3000 RSLCnano and Q-Exactive HF; Thermo Fisher Scientific) using a 200-min gradient. For this purpose, the peptides were sampled on a precolumn (300 µm × 5 mm PepMap C18; Thermo Fisher Scientific) and separated in a 75 µm × 250 mm C18 column (Reprosil-Pur 120 C18-AQ, 1.9 µm, Dr. Maisch). MS and MS/MS data were acquired by Xcalibur (Thermo Fisher Scientific).

Peptides and proteins were identified by Mascot (version 2.6.0, Matrix Science) through concomitant searches against the Uniprot database (*Homo sapiens* taxonomy, June 2019 download), a homemade classical database containing the sequences of classical contaminant proteins found in proteomic analyses (human keratins, bovine serum proteins, trypsin, etc.), and the corresponding reversed databases. Trypsin/P was chosen as the enzyme, and three missed cleavages were allowed. Precursor and fragment mass error tolerances were set at, respectively, at 10 ppm and 25 mmu. Peptide modifications allowed during the search were: carbamidomethyl (C, fixed), acetyl (protein N-term, variable), and oxidation (M, variable). The Proline software (Bouyssié et al, 2020) was used for the compilation, grouping, and filtering of the results (conservation of rank 1 peptides, peptide length ≥ 6 amino acids, peptide score ≥ 25, and false discovery rate of peptide-spectrum match identifications < 1% as calculated on peptide-spectrum match scores by employing the reverse database strategy). Proline was then used to perform a compilation, grouping and MS1 label-free quantification of the identified protein groups based on razor and specific peptides. Extracted abundances were normalized using variance stabilizing normalization in ProStaR (Wieczorek et al, 2017). Only proteins quantified with a minimum of five peptides were further considered. Proteins with a fold change ≥5 between 10 mM Glc and 0.1 mM Glc conditions were considered to have a decreased expression under glucose shortage. Among them, proteins with a fold change ≥2 between 0.1 mM Glc and 0.1 mM Glc + GlcNAc conditions were considered to have a partial surface localization rescue with GlcNAc.

### Plasmid transfection and luciferase assay

Hek293 cells were stably transfected with the plasmid pcDNA3-ER-LucT (donated by Dr. J Contessa) by lipofection using Lipofectamine 2000 reagent (Invitrogen), according to the manufacturer's instructions. Luciferase assays (E1910; Promega) were performed in Hek293 ER-LucT, and luciferase activities were measured with a microplate reader (Tecan, Infinite M200 PRO; Life Science).

### Flow cytometry analyses

HBEC-RL53 and H358 cells were washed in PBS containing 1% BSA and 0.1% NaN3 and incubated with saturating concentrations of the EGFR primary antibody (mca1784; Bio-Rad) for 30 min at 4°C. After fixation with paraformaldehyde 3.2%, they were incubated with the Alexa Fluor 647 (Invitrogen) secondary antibody for 30 min and then washed and resuspended in PBS. For apoptosis and cell death

analysis, AnnexinV-APC PI kit (BioLegend) was used according to the manufacturer's instructions. Samples were analyzed by flow cytometry using a FACSCanto II flow cytometer (BD Biosciences). The population of interest was gated according to its FSC/SSC criteria. Data were analyzed with FlowJo software (BD Biosciences). The representative experiment out of three independent experiments was shown.

## Statistical analysis

Statistical analyses were performed using the GraphPad Prism 8 software (GraphPad Software) via the Shapiro-Wilk normality test followed by parametric (paired or unpaired $t$ test) or nonparametric (Wilcoxon or Mann–Whitney) two-tailed test as indicated in figure legends. All data are expressed as means ± SEM of at least three independent experiments, ns, nonsignificant, *$P < 0.05$, **$P < 0.01$, ***$P < 0.001$, ****$P < 0.0001$.

# Supplementary Information

# Acknowledgments

We thank Brigitte Manship (Centre de Recherche en Cancérologie de Lyon) for reviewing the manuscript; Roxane Pommier (Centre de Recherche en Cancérologie de Lyon) for help with statistical analysis; Nicolas Gadot and Marie Brevet (Research Pathology Platform, Centre de Recherche en Cancérologie de Lyon) for their assistance in histological staining and TMA sample analysis; Joseph Contessa (Yale University Medical School) for providing the ER-LucT plasmid; Manon Nivet (Inserm U1242) and Joelle Fauvre (Centre de Recherche en Cancérologie de Lyon) for technical assistance. H Dragic is a recipient of a fellowship from la Ligue Nationale Contre le Cancer. This work was supported by the Institut National du Cancer (7981), la Ligue Nationale Contre le Cancer (R18159CC), and Rennes Métropole (R20167NN). The proteomic experiments were partially supported by Agence Nationale de la Recherche under projects ProFI (Proteomics French Infrastructure, ANR-10-INBS-08) and GRAL, a program from the Chemistry Biology Health Graduate School of University Grenoble Alpes (ANR-17-EURE-0003).

## Author Contributions

H Dragic: conceptualization, investigation, and methodology.
A Barthelaix: investigation and methodology.
C Duret: investigation.
S Le Goupil: investigation.
H Laprade: investigation.
S Martin: investigation.
S Brugiere: investigation.
Y Coute: investigation.
C Machon: investigation.
J Guitton: investigation.
J Rudewicz: investigation.
P Hofman: investigation and methodology.
S Lebecque: investigation and methodology.
C Chaveroux: investigation.
C Ferraro-Peyret: investigation.
T Renno: investigation and methodology.
SN Manie: conceptualization, supervision, investigation, methodology, and writing—original draft, review, and editing.

## Conflict of Interest Statement

The authors declare that they have no conflict of interest.

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
