## [Reviewer comments · Life Science Alliance]

Life Science Alliance

The Hexosamine Pathway and Coat Complex II promote malignant adaptation to nutrient scarcity

Helena Dragic, Audrey Barthelaix, Cedric Duret, Simon Le Goupil, Hadrien Laprade, Sophie Martin, Sabine Brugiere, Yohann Couté, Christelle Machon, Jérôme Guitton, Justine Rudewicz, Paul Hofman, Serge Lebecque, Cedric Chaveroux, Carole Ferraro-Peyret, Toufic Renno, and Serge Manie

DOI: <https://doi.org/10.26508/lsa.202101334>

Corresponding author(s): Serge Manie, Inserm U1242, Centre de Lutte Contre le Cancer Eugène Marquis, Université de Rennes

Review Timeline:

Submission Date:	2021-12-10
Editorial Decision:	2021-12-13
Revision Received:	2022-03-12
Editorial Decision:	2022-03-16
Revision Received:	2022-03-23
Accepted:	2022-03-24

Transaction Report:

Please note that the manuscript was previously reviewed at another journal and the reports were taken into account in the decision-making process at Life Science Alliance.

Referee #1 Review

Report for Author:

Dragic and others investigate the role of the hexosamine pathway in lung adenocarcinoma under conditions of glucose limitation. First, using TCGA database and by analysis of protein expression, they found that GFAT1 is increased in tumor cells of LUAD but not LUSC. Metabolite analysis also support that UDP-HexNAc levels are specifically increased in LUAD tumors. Using immortalized human bronchial epithelial cells (HBECs) with KRas mutation, they found increased GFAT1 mRNA and protein expression upon incubation in low glucose-containing media. Relative abundance of UDP-HexNAc under low glucose conditions was higher compared to UDP-glucose and GDP-Hexose. They then compared O-GlcNAcylation vs N-glycosylation under these conditions and using RL2 and CTD110.6 antibodies, they showed a dramatic decrease in expression of proteins recognized by RL-2 whereas there was an increase and distinct expression of proteins recognized by CTD110.6 antibody, suggesting that N-glycosylation, albeit at abnormal/truncated forms still occurs during glucose limitation. Next, they supplemented the glucose-limited culture with GlcNAc and found that this enhanced UDP-HexNAc levels, O-GlcNAcylation as well as possibly N-glycosylation as revealed by CTD110.6 immunoblotting and use of a luciferase reporter. In a proteomic analysis to determine expression of proteins that are restored upon GlcNAc supplementation, they found EGFR to be significantly restored and thus further analyzed its expression during glucose limitation. Flow cytometry confirmed that EGFR surface expression is restored upon GlcNAc supplementation of glucose-limited cells. Although full N-glycosylation is not achieved by GlcNAc supplementation, EGFR signaling is restored and cell viability is increased when analyzed in 3D culture. Analysis of human LUAD samples also revealed a correlation in expression of GFAT1 and EGFR. Lastly, they also found correlation in expression of COPII proteins in LUAD and that knockdown of a COPII protein decreased EGFR surface expression. Based on these findings, they concluded that the hexosamine pathway and COPII "promote malignant adaptation to nutrient scarcity."

The studies are interesting and support previous studies that highlight the metabolic reprogramming of the HBP in cancer particularly lung cancer. The data are also convincing and robust. A novelty in the current study would be the finding that HBP upregulation via GlcNAc supplementation under low glucose conditions could rescue EGFR expression and expression of N-glycosylated proteins (with truncated N-glycans) and cell viability. The distinct effect of supplementation on 2D vs 3D cultures is also interesting, However, the mechanisms are not fully addressed, hence the studies remain preliminary at this point. The effect of GlcNAc supplementation on the HBP and EGFR glycosylation would need to be addressed in more detail as it remains

unclear whether the effect of GlcNAc is due solely on HBP activation or via its metabolism that could enhance availability of glucose. How GlcNAc supplementation may also affect signaling that could explain EGFR upregulation and/or cell survival would need to be addressed. The involvement of COPII would also need further investigation as it is highly preliminary at this point and somewhat deviates from the main storyline.

Other specific comments:

1. Figure 3: The authors should also analyze the LUSC vs LUAD using RL2 and CTD antibodies.
2. Figure 3: Since cultures were grown for 48 hr, other nutrients could be limiting and could thus affect glycosylation (as well as translation). Authors should perform glucose withdrawal experiments at short time points to examine its effect. They should also use glycolytic inhibitors to verify requirement for glucose or glycolytic metabolites.
3. Figure 4: How GlcNAc fluxes through the HBP vs metabolized and be utilized to enter other metabolic pathways would need to be addressed.
4. Figure 4A: They should compare the levels of UDP-HexNAc in 10 mM vs .1 mM glucose +/- GlcNAc to provide insights on how GlcNAc is utilized under low vs high glucose.
5. Figure 4B: need to measure GDP-mannose to support that they remain low upon GlcNAc supplementation.
6. Figure 4D: Luciferase expression decreases significantly during glucose limitation, Hence, the GlcNAc supplementation may also affect translation, which could indirectly enhance expression of properly N-glycosylated proteins.
7. The use of CTD110 antibody to measure N-glycosylation is highly correlative. The authors should consider using mass spectrometry to assess the rescue of N-glycosylation during GlcNAc supplementation.
8. Fig 6A and D: GlcNAc supplementation does not fully rescue EGFR levels despite a better rescue of N-glycosylation (based on size) in 6D. The authors should address the mechanisms behind this. The effect of GlcNAc on EGFR transcription, translation, N-glycosylation should be closely interrogated. Furthermore, as mentioned above, it is not clear whether GlcNAc itself (through entry into HBP) is mediating this response.

Referee #2 Review

Report for Author:

Dragic et al. "The Hexosamine Pathway and Coat Complex II promote malignant adaptation to nutrient scarcity "

The authors study the effect of low glucose concentrations on a preclinical model of lung adenocarcinoma (LUAD) using human lung epithelial cells genetically manipulated to have abnormalities in TP53 and oncogenic KRAS. They assess metabolism in the hexosamine biosynthetic pathway (HBP) and find that such glucose "starvation" (which they feel mirrors that found in parts of human LUADs) leads to production of cell surface proteins bearing truncated N-glycans (they highlight that happening for EGFR). They show that upregulation of HBP and upregulation of COPII expression overcomes this alteration. They also perform a series of immunohistochemical studies of human LUADs showing alteration of expression of HBP components such as GFAT1 and its association with activation of wild-type EGFR in tumor samples. They also find that these changes are in LUADs but not found in lung squamous cancers (LUSCs). They also present selected confirmatory data using the non-small cell lung cancer (NSCLC) line H358. In addition, they find the HBP effect is greater in 3D compared to 2D culture. They conclude: "In conclusion, this work reveals that LUAD distinctive features provide protective functions to the malignant cells facing low glucose stress, strengthening the therapeutic interest in targeting the HBP for this disease. In addition, it raises questions for future investigations on the quality control process of N-glycosylation in the ER under stress. " No studies of mutant EGFR or studies showing the impact of targeting the HBP on tumor growth are presented.

Comments to the Authors:

1. The experiments are technically well done and clearly presented. The Discussion would have benefited from a summary figure describing the mechanism that occurs in LUADs would have helped the reader to capture their major conclusions.
2. The immediate question that arises is whether the findings are similar for LUADs with mutant EGFR compared to those with wild-type EGFR. The manuscript would have been greatly strengthened by such data no matter which way it comes out.
3. A major implication that the authors raise is that this alteration found in LUADs could be a therapeutic vulnerability. Thus, data on the effect of targeting the HBP pathway and its effect on LUAD growth in vitro and potentially in vivo (xenograft) would have greatly strengthened the manuscript.

Referee #3 Review

Report for Author:

This is a manuscript by Dragic et al. By integrating targeted metabolomics along with a series of biochemistry assays authors found that upon HBP upregulation can alleviate truncated N-glycan synthesis upon glucose limitation. The authors further noted that HBP activation via high levels of GFPT1 under low glucose promotes EGF stimulation of anchorage-independent growth

and is correlated with EGFR activation in LUAD tissues. Overall, the manuscript lacks scientific rigor and thus has multiple flaws/gaps in their rationale and it cannot be published in this journal unless they significantly fortify their logic. Major concerns regarding the manuscript are as shown below:

1. Authors treated cells with GlcNAc to activate hexosamine 'biosynthesis' pathway. GlcNAc is considered as a hexosamine 'salvage' pathway metabolite (PMID: 31272438 and other research articles), not the HBP metabolic intermediate mainly because it can bypass GFPT reaction, the rate-limiting step in the pathway. Authors need to pay attention to use the term 'HBP' when using GlcNAc.
2. Along with the Major concern 1, it is unclear how the authors jumped into the idea that under glucose limited conditions, HBP activation via GFPT1 promotes EGFR activation and tumorigenesis. All the experiments were performed with GlcNAc treatment, which is not related to GFPT (neither GFPT1 nor GFPT2) but associated with NAGK. It is unclear how they assayed the salvage pathway by adding GlcNAc and tried to connect the findings with GFPT1. Basically, they investigated the significance of salvage pathway then connected GFPT1 with their findings. Authors should have also checked NAGK activation under the glucose-limited conditions. This is the most concerning point in the manuscript.
3. Authors need to discuss discrepancy between previous findings (PMID: 29760045 and PMID: 33257855) and their finding that GFPT1, not GFPT2, expression is higher in LUAD. Additionally, GFPT1 may not be a driving gene to promote tumor aggressiveness even if its expression is higher in LUAD compared with normal tissues given that GFPT1 expression is not associated with poor prognosis in LUAD.
4. It is unclear how they normalized metabolomics data. While they used internal standard, they also need to normalize the data by tumor mass (mg of tissues) or protein quantification. Otherwise, the data is not convincing.
5. In Fig. 1C, GFPT1 expression seems higher both in LUSC and LUAD. Authors should provide quantification data before concluding that LUAD, but not LUSC, shows higher GFPT1 expression.
6. In Fig. 2B, GFPT2 expression should be measured as control to prove their assumption that GFPT1, but not GFPT2, is involved in HBP activation under glucose limited condition. In fact, the authors never provided any evidence that GFPT1 is the key/only gene in the HBP upregulated upon glucose deprivation.
7. Authors need to explain more in details about Figure 4. It is hard to understand why ER-LucT levels go down but signal increases dramatically. Additionally, the HBP flux needs to be investigated before making a conclusion that HBP flux impacts mannose flux by culturing cells with stable isotope labeled GlcNAc and check M6P, M1P, and GDP-mannose etc.
8. Authors never used genetic suppression to prove their findings. Tunicamycin experiments should be validated by silencing genes in N-glycosylation pathway.
9. They only used one cell line throughout the manuscript. It would be more convincing if they used at least one more cell line; there are multiple HBECs available.

December 13, 2021

Re: Life Science Alliance manuscript #LSA-2021-01334-T

Dr. Serge N Manie
Inserm U1242, Centre de Lutte Contre le Cancer Eugène Marquis, Université de Rennes
France

Dear Dr. Manie,

Thank you for submitting your manuscript entitled "The Hexosamine Pathway and Coat Complex II promote malignant adaptation to nutrient scarcity" to Life Science Alliance. We invite you to submit a revised manuscript addressing the following Reviewer comments:

- Address Reviewer 1's specific comments #1, 2 and 6.
- Address Reviewer 2's comment #1, and comment #2 if data is readily available.
- Address Reviewer 3's comments.

Thank you for this interesting contribution to Life Science Alliance. We are looking forward to receiving your revised manuscript.

Sincerely,

Eric Sawey, PhD
Executive Editor
Life Science Alliance
<http://www.lsa-journal.org>

- A letter addressing the reviewers' comments point by point.
- An editable version of the final text (.DOC or .DOCX) is needed for copyediting (no PDFs).
- High-resolution figure, supplementary figure and video files uploaded as individual files: See our detailed guidelines for preparing your production-ready images, <https://www.life-science-alliance.org/authors>
- Summary blurb (enter in submission system): A short text summarizing in a single sentence the study (max. 200 characters including spaces). This text is used in conjunction with the titles of papers, hence should be informative and complementary to the title and running title. It should describe the context and significance of the findings for a general readership; it should be written in the present tense and refer to the work in the third person. Author names should not be mentioned.
- By submitting a revision, you attest that you are aware of our payment policies found here: <https://www.life-science-alliance.org/copyright-license-fee>

B. MANUSCRIPT ORGANIZATION AND FORMATTING:

Dear Dr Sawey,

We thank the reviewers for their constructive comments. We hope these have been adequately addressed in our revised manuscript (changes are marked in red) and that the manuscript is now suitable for publication in Life Science Alliance.

Sincerely yours,
Serge Manié

Reviewer #1:

1. Figure 3: The authors should also analyze the LUSC vs LUAD using RL2 and CTD antibodies.

We have performed these analyses using original cellular lysates prepared for the experiments shown in Figure 1C. Of note, due to limited amount of sample available for one LUSC patient, we favored the CDT110 detection for it. RL2 signal intensity (O-GlcNAcylated proteins) was increased in 3 LUSC and 2 LUAD tumor tissues compared to adjacent normal tissue, and was not affected or decreased in 2 LUAD samples (#2 and #3, presented in Figure S2). The increased RL2 signal in a paired sample correlated with the respective amount of OGT that catalyzes O-GlcNAc modifications. These results indicate that O-GlcNAcylation in our tumor tissue samples is mainly driven by OGT amounts and cannot be simply interpreted as a readout of glucose availability or HBP activation. The staining pattern detected by the CDT110 antibody was less marked but followed the same trend as RL2 staining. It is conceivable that in a heterogeneous bulk tumor tissue, the relative high abundance of O-GlcNAcylated proteins (PMID: 22645316) conceals CDT110 cross-reaction with the less abundant N-GlcNAc2-modified proteins occurring in distinct cell clusters. This issue is cleared up when analyzing a cell population under controlled and homogeneous culture conditions in vitro. We have added this to the Results section of the revised manuscript (**text page 8** and **Figure S2**).

2. Figure 3: Since cultures were grown for 48 hr, other nutrients could be limiting and could thus affect glycosylation (as well as translation). Authors should perform glucose withdrawal experiments at short time points to examine its effect. They should also use glycolytic inhibitors to verify requirement for glucose or glycolytic metabolites.

We acknowledge the reviewer's concerns and indeed we have not stressed them enough in the manuscript. We found that 5.5 mM glucose was exhausted in cell cultures after 17h to 24h (Figure EV1A). Thus, we increased glucose concentration up to 10 mM in the control condition to avoid glucose shortage over 24h of culture and, importantly, all media used were refreshed daily. Beside glucose concentration, all media are identically formulated. It is reasonable to consider that if depletion of one component other than glucose is significantly implicated in hypoglycosylation or reduced cellular proliferation, we should see its effect, irrespective of the

media used. To further document this point, we have performed the suggested complementary experiment using 2-Deoxy-D-glucose (2-DG), which competes with the metabolism of glucose (PMID: 31905745). 2-DG efficiently reduced EGFR N-glycosylation in the presence of 10mM glucose, and this effect was partly counteracted by GlcNAc supplementation. These results support a requirement for glucose in EGFR glycosylation rather than a limitation in another nutrient. This has been added in the revised manuscript (**text page 11** and **Figure S3B**).

6. Figure 4D: Luciferase expression decreases significantly during glucose limitation, Hence, the GlcNAc supplementation may also affect translation, which could indirectly enhance expression of properly N-glycosylated proteins.

The reviewer raises an important question that we have now examined by assessing the effect of GlcNAc on the N-glycosylation of the transmembrane protein PD-L1, because its translation rate should not be affected by glucose limitation-induced ER stress (PMID: 32905506). While EGFR amount was reduced in low glucose conditions, PD-L1 level was much less affected (presented in Figure S4A). Nonetheless, glucose shortage induced a smear of PD-L1 bands toward the 35 kDa region, indicating an altered PD-L1 glycosylation (PMID: 27572267). Under this condition, GlcNAc supplementation limited both the extent of PD-L1 smear and EGFR hypoglycosylation. These results indicate that GlcNAc supplementation can restore the glycosylation of proteins upon low glucose irrespective of their translation rate in the ER. This has been added in the revised manuscript (**text page 9, 12** and **Figure S4A**).

Reviewer #2:

1. The experiments are technically well done and clearly presented. The Discussion would have benefited from a summary figure describing the mechanism that occurs in LUADs would have helped the reader to capture their major conclusions.

We thank the reviewer for the positive comments. As suggested, we now provide a schematic of the mechanisms that occur in LUAD (**page 18** and **Figure 9**).

2. The immediate question that arises is whether the findings are similar for LUADs with mutant EGFR compared to those with wild-type EGFR. The manuscript would have been greatly strengthened by such data no matter which way it comes out.

The reviewer raises an excellent point. To address it, we tested glucose limitation with or without GlcNAc supplementation in PC9 human adenocarcinoma cell line harboring *EGFR*-activating mutation (delE746-A750) and in HBECS expressing *EGFR-L858R* mutant instead of *Krasv12* oncogene. Similar to wild-type EGFR, N-glycosylation of both EGFR mutants was reduced by glucose shortage and restored upon GlcNAc supplementation (presented in Figure S4C). However, GlcNAc supplementation sustained the autophosphorylation of EGFR mutants in HBECS but not in PC9 cells. In addition, phosphorylation of the EGFR downstream targets ERK

and Gab1 were not affected by exogenous GlcNAc in either cell lines. While further analysis would be required to reach definitive conclusions, these results suggest that the sustained production of UDP-GlcNAc upon low glucose does not strongly affect the oncogenic signaling of EGFR mutants. A possible reason might be that in contrast to wild-type EGFR, mutant EGFR-dependent signaling is constitutively activated and does not depend on the binding of an activating ligand at the cell surface (PMID: 20388509). This has been added in the revised manuscript (**text page 12 and 16 and Figure S4C**).

Reviewer #3:

1. Authors treated cells with GlcNAc to activate hexosamine 'biosynthesis' pathway. GlcNAc is considered as a hexosamine 'salvage' pathway metabolite (PMID: 31272438 and other research articles), not the HBP metabolic intermediate mainly because it can bypass GFPT reaction, the rate-limiting step in the pathway. Authors need to pay attention to use the term 'HBP' when using GlcNAc.

We agree with the reviewer's comment, this point merits clarification. We have used exogenous GlcNAc as a means to expand the UDP-GlcNAc pool in a cellular context where glucose scarcity impedes its production. The rationale was to better decipher the outcomes of an increase in UDP-GlcNAc *per se* on the elongation of N-linked structures in the ER and on EGFR biology. Even though this approach mimics a robust upregulation of HBP's terminal metabolite, it is not to be considered as an endogenous GFPT-dependent HBP flux whose extent is limited by glucose availability *in vitro*. This clarification has now been made throughout the revised manuscript (**text pages 8, 9, 10, 11, 12, 16, 18**).

2. Along with the Major concern 1, it is unclear how the authors jumped into the idea that under glucose limited conditions, HBP activation via GFPT1 promotes EGFR activation and tumorigenesis. All the experiments were performed with GlcNAc treatment, which is not related to GFPT (neither GFPT1 nor GFPT2) but associated with NAGK. It is unclear how they assayed the salvage pathway by adding GlcNAc and tried to connect the findings with GFPT1. Basically, they investigated the significance of salvage pathway then connected GFPT1 with their findings. Authors should have also checked NAGK activation under the glucose-limited conditions. This is the most concerning point in the manuscript.

We found that GFPT1 was upregulated in both LUAD tissues and LUAD cell lines subjected to low glucose. That GFPT1 may contribute to the sustained UDP-GlcNAc production *in vivo* was suggested by the positive association between its expression level and EGFR activation in LUAD tissues (figure 7). At difference with LUAD tissues, UDP-GlcNAc levels in cell lines were not increased but better maintained than those of other glucose-dependent nucleotides sugars (Figure 2C). This selective preservation of UDP-GlcNAc amounts, in the absence of exogenous GlcNAc supplementation, was used to fuel the N-glycosylation pathway (figure 3) and to partly

compensate for the reduced EGFR surface expression (figure 5C). The purpose of using GlcNAc supplementation next was to demonstrate that a further increase in UDP-GlcNAc pools under glucose-limited condition could fully restore the elongation of N-glycan in the ER and EGFR surface expression (figures 4, 5 and 6), and we did not connect it to the increase in GFPT1. How cells in culture succeeded in maintaining elevated levels of endogenous UDP-GlcNAc has not been investigated as we have focused our study on the cellular use of the residual HBP terminal metabolite when glucose becomes scarce, because very little was known about it. To make this point clearer, the sentence in the discussion section on page 17, has been modified as follows (bolded text): “*Better preserved UDP-GlcNAc amounts may result from the increased channeling of traces of glucose-derived metabolites into the HBP **via the upregulated rate-limiting enzyme GFPT1** and/or from GlcNAc salvage from recycled glycoconjugates **via the N-acetylglucosamine kinase (NAGK)**”.*

To address the referee comment, we have examined the level of NAGK under the glucose-limited conditions. We found that NAGK mRNA expression was decreased by glucose scarcity and that its protein amounts were virtually unaffected, indicating that NAGK expression is not regulated by low glucose in this cellular model (presented in Figure S6). Yet, NAGK activity might be regulated through post-translational modifications, as recently suggested (PMID: 34844667), but very little is currently known about such a regulation. This has been added to the Discussion section (**text page 17 and Figure S6**).

3. Authors need to discuss discrepancy between previous findings (PMID: 29760045 and PMID: 33257855) and their finding that GFPT1, not GFPT2, expression is higher in LUAD. Additionally, GFPT1 may not be a driving gene to promote tumor aggressiveness even if its expression is higher in LUAD compared with normal tissues given that GFPT1 expression is not associated with poor prognosis in LUAD.

The apparent discrepancy between these previous findings and our work with regard to GFPT1 expression can be readily explained. First, Zhang et al. (PMID: 29760045) showed that LUAD malignant cells express significantly more GFPT1 than GFPT2, and that GFPT2 is in fact largely expressed by cancer-associated fibroblasts (figure 3C of their manuscript). Given that malignant cells are more abundant than CAFs in tumor samples, our analysis of bulk tumor gene expression revealed an expected preponderant GFPT1 increase. Identical trends are observed when using web-based tools, such as UAFLAN (see #1). Second, the work by Kim et al. (PMID: 33257855) demonstrated that the GFPT2 paralog plays an important role in the tumorigenesis of the LUAD subtype containing concurrent mutations in KRAS and LKB. This function for GFPT2 was not conserved in cells with oncogenic KRAS only, the most frequent oncogenic aberration in LUAD. Hence, the selective dependence on GFPT2 in the KL subtype that accounts for 6-12% of LUAD, should not be extended to other LUAD subtypes.

As noted by the referee, GFPT1 expression alone was not found associated with poor prognosis in LUAD (PMID: 33257855). In this study, the authors used the online bioinformatic tool KM plotter. However, the use of Easysurv, another web-based tool that can perform advanced survival analyses using data from the TCGA, showed that GFPT1 is associated with poor prognosis, predominantly in stages I and II of the disease (see #2, we used the univariate analysis and selected the optimal cutoff to generate a Kaplan-Meier plot of GFPT1). Given that each tool has its own strength, we would argue that the prognostic value of GPT1 expression in LUAD requires further investigation.

#1 <http://ualcan.path.uab.edu/cgi-bin/TCGAExResultNew2.pl?genenam=GFPT1,GFPT2&ctype=LUAD>

#2 <https://easysurv.net/#/app/result/general/detail/2726>).

4. It is unclear how they normalized metabolomics data. While they used internal standard, they also need to normalize the data by tumor mass (mg of tissues) or protein quantification. Otherwise, the data is not convincing.

The data were actually normalized to the amount of protein in each sample as indicated in the Materials and Methods section, page 22: “Metabolites were extracted using cold methanol/water mixture (...) Supernatants were frozen using liquid nitrogen and sent for LC/MS-MS analysis (HCL, Lyon-Sud). Precipitated proteins were solubilized in RIPA containing 8 M urea and quantified using the Lowry method. A volume of lysate corresponding to 0.5-0.6 µg of proteins was used for each sample and the labelled internal standard (CTP_{13C}) solution was added”. The addition of cold methanol to samples significantly lowers the solubility of the proteins, resulting in proteins precipitating out. The precipitates were removed by centrifugation for quantification, and the supernatants used for metabolomic analyses

5. In Fig. 1C, GFPT1 expression seems higher both in LUSC and LUAD. Authors should provide quantification data before concluding that LUAD, but not LUSC, shows higher GFPT1 expression.

GFPT1 mRNA levels were clearly higher in LUAD (Figure 1B). While both LUAD and LUSC showed increased amounts in GFAT1 protein, quantification of the blots confirmed that this increase was twice higher in LUAD. This has been added to the Results section page 5 (**Table S1**).

6. In Fig. 2B, GFPT2 expression should be measured as control to prove their assumption that GFPT1, but not GFPT2, is involved in HBP activation under glucose limited condition. In fact, the authors never provided any evidence that GFPT1 is the key/only gene in the HBP upregulated upon glucose deprivation.

Because GFPT1, and not GFPT2, was upregulated in the bulk analysis of LUAD tissues (see also response to referee's comment #3 above), we looked for its expression in cell lines upon glucose scarcity. This rate-limiting enzyme was indeed increased, thus mirroring the *in vivo* situation and pointing toward a role for the HBP in cells facing detrimental conditions. However, we did not aim to exclude other actors of the HBP that may also be upregulated in cell cultures, such as GFPT2. Yet, we appreciate the referee's comment and have examined the level of GFPT2 under glucose-limited conditions. We found that GFPT2 mRNA expression was also increased in HBECs under glucose scarcity but at a very low level compared to GFPT1. This result has been added to the Results section (**text page 6** and **Figure S1C**).

7. Authors need to explain more in details about Figure 4. It is hard to understand why ER-LucT levels go down but signal increases dramatically. Additionally, the HBP flux needs to be investigated before making a conclusion that HBP flux impacts mannose flux by culturing cells with stable isotope labeled GlcNAc and check M6P, M1P, and GDP-mannose etc.

The bioluminescence of the ER-LucT construct is impaired when the protein becomes N-glycosylated in the ER (PMID: 20413434). This N-glycosylation induces a shift of the protein towards higher molecular weights on Western blots. Figure 4 shows that after 48h of glucose shortage, ER-LucT molecular weight was reduced, indicating an impaired N-Glycosylation. As expected, the reduced N-glycosylation translated into an increase in bioluminescence. Under this condition however, the quantity of ER-LucT was also decreased, likely because of low glucose-induced ER stress that reduces global translation (PMID: 11106749). Indeed, the reduced expression of ER-LucT was restored by exogenous GlcNAc that relieves low glucose-induced ER stress (PMID:30305738; 23868065; 23395000). We have added this clarification to the Results section of the revised manuscript (**text page 9**).

It was not our intention to conclude that HBP flux impacts mannose flux. How GlcNAc supplementation relieves the impaired mannose addition during N-glycan building is currently unclear as stated in page 10 and 18. It may result from a restored mannose flux as mentioned by the referee, or alternatively, from the unlocking of a low glucose-impaired usage of residual mannose-derived metabolites. Ongoing experiments should help to answer this question, but they will require too much time to be completed for inclusion in the present study. We hope the referee agrees that this can be considered an area for future pursuit.

8. Authors never used genetic suppression to prove their findings. Tunicamycin experiments should be validated by silencing genes in N-glycosylation pathway.

We initially tried to target DPAGT1 that catalyzes the branching of the first GlcNAc residue onto Dol-P. Unfortunately, the silencing of DPAGT1 proved to be toxic in HBECs (see attached results below). Thus, we turned to tunicamycin at 0.5 µg/mL, a low dose found in preliminary experiments to limit its cytotoxic potential in HBECs.

DPAGT1 silencing. (A) mRNA levels of DPAGT1 in HBECs grown for 72 h in complete KSFM and treated with the indicated siRNAs (SMART-pool, Dharmacon). **** $p < 0.0001$ (un-paired T-test, $n = 2$ independent experiments, mean \pm SEM). (B) IncuCyte relative confluency of HBECs, 72 h post DPAGT1 siRNA treatment. **** $p < 0.0001$ (un-paired T-test, $n = 2$ independent experiments, mean \pm SEM).

9. They only used one cell line throughout the manuscript. It would be more convincing if they used at least one more cell line; there are multiple HBECs available.

Actually, the main findings were indeed recapitulated in a different cell line, the H358 LUAD cell line harboring also a *KRAS* mutation. It was presented in pages 12,13 and 15, and Figures S4B, 4F and S5C.

March 16, 2022

RE: Life Science Alliance Manuscript #LSA-2021-01334-TR

Dr. Serge N Manie
Inserm U1242, Centre de Lutte Contre le Cancer Eugène Marquis, Université de Rennes
Avenue de la bataille Flandres-Dunkerque
Rennes 35042
France

Dear Dr. Manie,

Thank you for submitting your revised manuscript entitled "The Hexosamine Pathway and Coat Complex II promote malignant adaptation to nutrient scarcity". We would be happy to publish your paper in Life Science Alliance pending final revisions necessary to meet our formatting guidelines.

- please make sure the author order in your manuscript and our system match
- please add the Twitter handle of your host institute/organization as well as your own or/and one of the authors in our system
- please add a legend for Table S2 to the main manuscript and provide all Tables in excel format
- please add a callout for Figure S3A to your main manuscript text

A. FINAL FILES:

B. MANUSCRIPT ORGANIZATION AND FORMATTING:

Sincerely,

March 24, 2022

RE: Life Science Alliance Manuscript #LSA-2021-01334-TRR

Dr. Serge N Manie
Inserm U1242, Centre de Lutte Contre le Cancer Eugène Marquis
Université de Rennes
Avenue de la bataille Flandres-Dunkerque
Rennes 35042
France

Dear Dr. Manie,

Thank you for submitting your Research Article entitled "The Hexosamine Pathway and Coat Complex II promote malignant adaptation to nutrient scarcity". It is a pleasure to let you know that your manuscript is now accepted for publication in Life Science Alliance. Congratulations on this interesting work.

DISTRIBUTION OF MATERIALS:

Again, congratulations on a very nice paper. I hope you found the review process to be constructive and are pleased with how the manuscript was handled editorially. We look forward to future exciting submissions from your lab.

Sincerely,
